# CrossQ: Task-Aligned Cross-Token Conditional Quantization for Late Interaction Retrieval

**Rohit Kumar Salla** [1]   **Manoj Saravanan** [1]   **Ramya Manasa Amancherla** [2]

## Abstract

Late-interaction retrievers like ColBERT achieve high quality but suffer from large multi-vector indices. Standard compression minimizes token reconstruction error, while ranking depends critically on preserving scores of sparse "winner" tokens. We introduce CrossQ, which adaptively improves effective token fidelity within documents by conditioning token codes on lightweight document context computed at indexing time (but not stored). CrossQ is trained with ranking-aligned objectives that preserve candidate score distributions and protect hard-negative margins. At 2 B/token, CrossQ improves MRR@10 by +0.010 over the strongest strictly footprint-matched quantization baseline and by +0.012 over the strongest candidate-matched system reference. On the nine-dataset BEIR subset reported in Appendix G.1, CrossQ improves average nDCG@10 by +0.009 at 4 B/token over the strongest candidate-matched system reference. At 4 B/token, CrossQ achieves $64\times$ raw token-storage reduction, approximately $61\times$ including metadata and approximately $58\times$ under conservative padding/alignment accounting. At 8 B/token, CrossQ + light fine-tuning retains approximately 98% of full-precision ColBERT MRR@10, improving the footprint-quality tradeoff for memory-constrained late-interaction retrieval.

## 1. Introduction

Neural retrieval is judged by *ranking*: for each query, do relevant documents appear in the top-$k$? Late-interaction retrievers such as ColBERT score documents by matching each query token to its best-matching document token (Khattab & Zaharia, 2020). This improves effectiveness but is costly: multi-vector indices dominate memory and max-

sim comparisons dominate query-time compute (Santhanam et al., 2022a), motivating compression of *document-side* token embeddings.

The max operator selects sparse *winner* tokens (Khattab & Zaharia, 2020); small perturbations on frequent winners can flip tight margins between a positive and a hard negative, while similar error on non-winners may not matter. Thus objectives that minimize average distortion (e.g., token MSE) can preserve embeddings on average yet distort the *relative score structure* that determines top-$k$ ranking.

**CrossQ: Cross-Token Conditional Quantization.** We propose **CrossQ**, an additive multi-codebook quantizer whose code selection is conditioned on a lightweight, permutation-invariant document context $h = f_{\mathrm{ctx}}(D)$ (Zaheer et al., 2017). For each token embedding $d_j$, we form a conditional representation $z_j = g(d_j, h)$ and select one code per codebook:

$$\hat{d}_j = \sum_{m=1}^{M} E_{c_{j,m}}^{(m)}, \qquad c_{j,m} = \arg\max_k \ell_{j,m,k}, \quad (1)$$

where $\ell_{j,m,k}$ are logits over the $K$ codewords in codebook $m$ (defined formally in Eq. 13). Context conditioning enables within-document effective-fidelity allocation: we compute h once at indexing time to choose codes, but store only discrete token codes (and shared codebooks), so the footprint remains fixed by the bytes/token budget.

Unlike residual quantizers, CrossQ uses a *parallel* additive structure: stage selections do not depend on prior-stage residuals.

Rather than optimize reconstruction, CrossQ is trained to preserve score structure using listwise distillation (Hinton et al., 2015),

$$\mathcal{L}_{\mathrm{list}}(Q) = \mathrm{KL}(\mathrm{softmax}(s_T/\tau) \,\|\, \mathrm{softmax}(s_S/\tau)), \quad (2)$$

together with a hard-negative pairwise ranking loss (Burges et al., 2005) (and optional light fine-tuning) to stabilize margins under aggressive compression. It is important to note that CrossQ is specifically designed for **max-sim late interaction**; extending this approach to other scoring operators (e.g., dot product) requires training signals aligned with their specific geometry.

---

[1]Department of Electrical and Computer Engineering, Virginia Tech, Blacksburg, VA, USA [2]Columbia University, New York, NY, USA. Correspondence to: Rohit Kumar Salla <rohits25@vt.edu>.

*Proceedings of the 43rd International Conference on Machine Learning*, Seoul, South Korea. PMLR 306, 2026. Copyright 2026 by the author(s).

**Contributions.**

- Average reconstruction error is a weak proxy for late-interaction ranking under max-sim.
- **CrossQ** conditions token codes on document context to allocate effective representational fidelity to retrieval-critical tokens at fixed bytes/token.
- Score-level listwise distillation plus hard-negative supervision improves ranking preservation, especially under domain shift.
- **Results:** On MS MARCO (Bajaj et al., 2018) and BEIR (Thakur et al., 2021), CrossQ improves footprint-matched MRR/Recall/nDCG and yields a better footprint–quality trade-off than strong baselines.

## 2. Background and Motivation

Late-interaction retrievers preserve token-level evidence but are expensive at scale: each document stores many contextual token embeddings, creating large multi-vector indices and costly max-sim scoring (Khattab & Zaharia, 2020; Santhanam et al., 2022b). We focus on *document-side compression* and explain why standard distortion objectives are misaligned with late-interaction ranking.

**Late interaction and winner tokens.** Queries and documents are encoded as token embeddings $Q = \{q_i\}_{i=1}^m$ and $D = \{d_j\}_{j=1}^n$ and scored by

$$s(Q, D) = \sum_{i=1}^m \max_{j \in [n]} \langle q_i, d_j \rangle, \qquad (3)$$

so only the maximizing ("winner") token contributes for each $q_i$ (Khattab & Zaharia, 2020). This yields strong evidence matching but makes rankings sensitive to winner changes and near-tie margin flips.

**Why reconstruction objectives miss the target.** PQ/OPQ-style compression minimizes average reconstruction error (Jégou et al., 2011; Ge et al., 2013; Johnson et al., 2021). Under max-sim, score mass concentrates on a small, query-dependent set of winners (Khattab & Zaharia, 2020; Santhanam et al., 2022a): small perturbations on frequent winners can flip a positive vs. hard negative, even when overall token distortion is low. Thus token MSE is a weak proxy for preserving top-$k$ ranking.

Token-wise quantization treats each $d_j$ independently. Documents mix redundant tokens with rare, high-salience evidence and late interaction amplifies this imbalance (Khattab & Zaharia, 2020). Although every token stores the same $M \log_2 K$ bits, uniform *effective precision* over-spends on redundant tokens and under-represents likely winners; the key challenge is *within-document allocation of effective precision* under a fixed storage budget.

**Relationship to conditional quantizers.** Recent learned quantizers condition code selection on local structure:

*Table 1.* Comparison with related compression methods.

| | QINCo2 | JPQ/RepCONC | Routing | CrossQ |
|---|---|---|---|---|
| Conditions on | Token resid. | Joint w/ enc. | Centroid dist. | Doc context |
| Scope | Per-token | Per-vector | Per-token | Cross-token |
| Target operator | Recon. (ANN) | Single-vec dot | Recon. (ANN) | Max-sim |
| Training loss | Recon. | Contrastive | Recon. | Ranking |
| Stored | Codes | Codes+enc. | Codes+routes | Codes only |

QINCo2 (Vallaeys et al., 2025) conditions on token residuals, routing methods condition on centroid distances. Both operate per-token independently and optimize reconstruction. Table 1 summarizes key differences. CrossQ conditions on *document-level* context, enabling cross-token capacity allocation and trains with ranking-aligned losses targeting max-sim directly. Crucially, the context $h(D)$ is discarded after indexing only integer codes are stored.

**Retrieval-oriented learned compression.** JPQ (Zhan et al., 2021) and RepCONC (Zhan et al., 2022) train product quantization end-to-end with the retriever, jointly optimizing encoder and codebooks under contrastive losses. Both target *single-vector* dense retrieval, where a document is represented by one pooled embedding. Extending these methods to multi-vector late interaction is not straightforward: max-sim depends on sparse winner tokens and cross-token interactions rather than a single pooled embedding, so per-token codebooks must preserve fine-grained score structure rather than coarse pooled similarity. Our baseline set therefore focuses on methods naturally applicable to multi-vector indices under matched bytes/token.

**Distillation for retrieval.** Teacher–student distillation has been used to compress retrievers (Xiao et al., 2022), typically transferring knowledge from cross-encoders to dense retrievers via score or representation matching. This is orthogonal to CrossQ and complementary: stronger teachers can feed our listwise score distillation objective. Our contribution is not distillation per se, but distillation *aligned to max-sim ranking structure* (winner-token sparsity, near-tie margin sensitivity) combined with document-conditioned code selection under strict bytes/token budgets.

## 3. Problem Setup

We study *late-interaction* retrieval under a strict index-memory budget. A trained late-interaction retriever encodes each document into contextual token embeddings. Storing these embeddings at full precision typically dominates index size and memory bandwidth during scoring. Our goal is to *compress only the document-side token embeddings* so that, at a fixed bytes/token budget, the compressed index preserves the *query-conditioned rankings* produced by the full-precision teacher.

**Setup and Notation** A query is $Q = \{q_i\}_{i=1}^m$ and a document is $D = \{d_j\}_{j=1}^n$ with $q_i, d_j \in \mathbb{R}^d$. We compress document tokens only (query tokens remain full precision), so any quality change is attributable to the document represen-

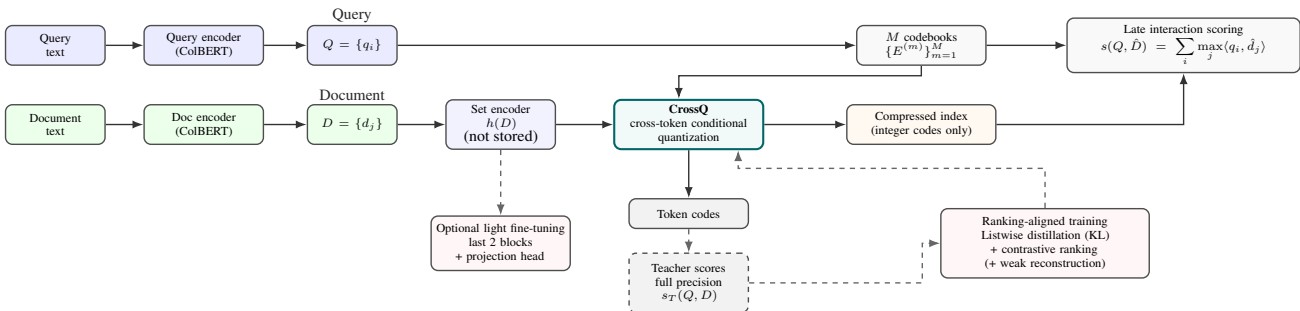

*Figure 1.* **CrossQ overview.** CrossQ conditions code selection on document-level context (computed at indexing time but *not stored*) and trains with ranking-aligned objectives to preserve score orderings under max-sim late-interaction scoring (Khattab & Zaharia, 2020). The deployed index stores only integer token codes, preserving standard late-interaction efficiency. Detailed mechanics of additive codebook decoding, the conditional code selector at each stage and query-time lookup-table scoring are given in Figs. 5–7.

tation. We write $\mathcal{D}$ for the corpus and $\hat{\mathcal{D}}$ for its compressed index.

Each document token embedding is mapped to a discrete code and decoded back to a reconstructed embedding:

$$\mathbf{c}_j = \mathcal{E}(d_j), \qquad \hat{d}_j = \mathcal{D}(\mathbf{c}_j), \qquad \hat{D} = \{\hat{d}_j\}_{j=1}^n.$$

In this work, $\mathcal{E}/\mathcal{D}$ implement additive multi-codebook quantization with $M$ codebooks and $K$ entries per codebook. Each token stores $M$ code indices (one per codebook), so the dominant storage cost is

$$\text{bits/token} \approx M \log_2 K,$$

plus small per-document metadata (e.g., offsets and lengths). We use $\tau$ for the listwise distillation temperature, $\gamma$ for pairwise-loss sharpness, $\tau_q$ for the quantizer softmax temperature used during soft-to-hard training, loss weights $(\alpha, \beta, \lambda)$ and context dimension $d_c$ (for CrossQ variants that use document context).

**Late-interaction scoring and winner tokens.** ColBERT-style late interaction scores a query–document pair by summing per-query-token maximum similarities (Khattab & Zaharia, 2020):

$$s(Q, D) = \sum_{i=1}^m \max_{j \in [n]} \langle q_i, d_j \rangle. \tag{4}$$

For each query token $q_i$, let $\pi(i; Q, D) \in [n]$ be an index attaining the maximum (ties broken arbitrarily). Then $d_{\pi(i;Q,D)}$ is the winner token for $q_i$ and we can rewrite

$$s(Q, D) = \sum_{i=1}^m \langle q_i, d_{\pi(i;Q,D)} \rangle.$$

This winner-take-all structure is what makes compression difficult: rankings can change either because (i) compression changes the winner identity for some $q_i$, or (ii) winners remain the same but compression noise alters *tight* score gaps between candidates (near ties).

At inference time we apply the same late-interaction operator to reconstructed document tokens:

$$\hat{s}(Q, D) \triangleq s(Q, \hat{D}) = \sum_{i=1}^m \max_{j \in [n]} \langle q_i, \hat{d}_j \rangle. \tag{5}$$

Our objective is to preserve the ranking induced by $\hat{s}(Q, \cdot)$ relative to $s(Q, \cdot)$, subject to a strict index footprint constraint. Equivalently, for each query we want the compressed index to return the same (or nearly the same) top-$k$ set as the teacher and to preserve the relative ordering among hard negatives that compete for the top ranks.

### 3.1. Budgeted Retrieval Problem

Let $\mathcal{Q}$ be the set of training queries. For each $Q \in \mathcal{Q}$ we assume a labeled positive document $D^+$ and a set of negatives $\mathcal{N}(Q)$ (e.g., in-batch negatives, mined hard negatives, or teacher-proposed candidates). We define the candidate set

$$\mathcal{C}(Q) = \{D^+\} \cup \mathcal{N}(Q),$$

and evaluate with standard retrieval metrics (MRR@k, Recall@k, nDCG@k). Importantly, the candidate set is where *ranking errors happen*: hard negatives in $\mathcal{N}(Q)$ often have scores close to $D^+$, so small score perturbations from compression can change their relative order.

Let $\hat{\mathcal{D}}$ denote the compressed corpus representation. We require

$$\text{Size}(\hat{\mathcal{D}}) \le B, \tag{6}$$

where $\text{Size}(\cdot)$ counts code storage plus small per-document metadata (e.g., offsets/lengths and alignment). In practice, we choose $(M, K)$ (and any fixed metadata scheme) to satisfy the target bytes/token budget, then train the compressor within that fixed footprint.

**Margins and near-tie sensitivity.** For each negative $D^- \in \mathcal{N}(Q)$, define teacher and student margins:

$$\Delta_T(Q; D^+, D^-) \triangleq s(Q, D^+) - s(Q, D^-), \tag{7}$$

$$\Delta_S(Q; D^+, D^-) \triangleq \hat{s}(Q, D^+) - \hat{s}(Q, D^-). \tag{8}$$

Ranking preservation requires $\text{sign}(\Delta_S) = \text{sign}(\Delta_T)$ for hard negatives. Empirically, most ranking errors are concentrated on *small* $\Delta_T$ (near ties), which motivates training signals that preserve not only the ordering but also the *relative gaps* between candidates.

# 4. Method: Cross-Token Conditional Quantization

CrossQ compresses a document $D = \{d_j\}_{j=1}^n$ *without treating tokens independently*. The design follows directly from late interaction: only a small, query-dependent subset of document tokens become max-sim winners, so uniform effective precision across all tokens is inefficient even when nominal storage is uniform. CrossQ addresses this mismatch by computing a lightweight, query-independent document context $h(D)$ at indexing time and conditioning each token's code selection on this shared signal. Crucially, *CrossQ stores only integer token codes in the index*. The context $h(D)$ is used to choose codes but is *not stored*, so the deployed footprint remains standard additive codes.

## 4.1. Additive Multi-Codebook Quantization

We represent each token using $M$ additive codebooks $\{E^{(m)}\}_{m=1}^M$ with $K$ entries each. Token $d_j$ is encoded as indices $\{c_{j,m}\}_{m=1}^M$ with $c_{j,m} \in [K]$ and reconstructed as

$$\hat{d}_j \triangleq \sum_{m=1}^M E_{c_{j,m}}^{(m)}. \tag{9}$$

This additive structure is attractive for late interaction: query-time dot products decompose into $M$ small table lookups, preserving fast max-sim execution. CrossQ differs from reconstruction-driven baselines in *how* it chooses the codes $c_{j,m}$: we learn a conditional assignment policy that targets ranking preservation under the max-sim operator rather than average reconstruction error.

## 4.2. Document Context via a Set Encoder

CrossQ computes a compact document context $h(D) \in \mathbb{R}^{d_c}$ that summarizes the document's global content. We use a permutation-invariant set encoder (DeepSets-style), treating the document as a set of token embeddings:

$$u_j = \phi(d_j) \in \mathbb{R}^{d_c}, \tag{10}$$

$$h(D) = \rho\left(\frac{1}{n}\sum_{j=1}^n u_j\right), \tag{11}$$

(Zaheer et al., 2017) where $\phi$ and $\rho$ are small MLPs. Intuitively, $h(D)$ captures coarse semantic structure (e.g., topic/style) that helps the quantizer decide which tokens should receive higher effective fidelity under a fixed bytes/token budget. We compute $h(D)$ at indexing time and use it only for code selection; it is *not stored* in the final index.

## 4.3. Conditional Code Selection

Given a token embedding $d_j$ and the document context $h(D)$, we form a conditional token representation

$$z_j \triangleq g([d_j; h(D)]), \tag{12}$$

where $[\cdot;\cdot]$ denotes concatenation and $g$ is a small MLP. For each codebook stage $m$, we compute logits over entries $k \in [K]$ via

$$\ell_{j,m,k} = \left\langle W^{(m)} z_j, \tilde{E}_k^{(m)} \right\rangle, \qquad k \in [K], \tag{13}$$

where $W^{(m)}$ is a learned projection and $\tilde{E}^{(m)}$ are *assignment keys*. We distinguish assignment keys $\tilde{E}^{(m)}$ from reconstruction values $E^{(m)}$: the keys parameterize assignment geometry, while the values define the stored codebooks used at inference. (When tied, $\tilde{E}^{(m)} \equiv E^{(m)}$.)

During training we use a temperature-controlled softmax:

$$\pi_{j,m,k} = \frac{\exp(\ell_{j,m,k}/\tau_q)}{\sum_{k'=1}^K \exp(\ell_{j,m,k'}/\tau_q)}. \tag{14}$$

Soft assignments make the quantizer differentiable and allow gradients to flow into the assignment network and codebooks. We decode using the expected codebook values:

$$\hat{d}_j = \sum_{m=1}^M \sum_{k=1}^K \pi_{j,m,k} E_k^{(m)}. \tag{15}$$

At indexing time, we store hard codes $c_{j,m} = \arg\max_k \pi_{j,m,k}$ for each token and stage.

## 4.4. Efficient Late-Interaction Scoring

Because CrossQ stores additive codes, token similarities decompose as

$$\langle q_i, \hat{d}_j \rangle = \sum_{m=1}^M \left\langle q_i, E_{c_{j,m}}^{(m)} \right\rangle. \tag{16}$$

For each query token $q_i$, we precompute lookup tables $T_i^{(m)}[k] = \langle q_i, E_k^{(m)} \rangle$ once per query. We then score a document using the standard max-sim late interaction, but operating on integer codes:

$$\hat{s}(Q, D) = \sum_{i=1}^m \max_{j \in [n]} \sum_{m=1}^M T_i^{(m)}[c_{j,m}]. \tag{17}$$

Thus, CrossQ preserves the late-interaction scoring structure while replacing stored floating-point token embeddings with compact integer codes and uses cross-token conditioning only to *choose* those codes at indexing time.

# 5. Task-Aligned Objective and Training

We train CrossQ with *ranking-aligned* supervision rather than pure reconstruction. The guiding principle is to optimize the same quantities that determine retrieval under

max-sim scoring: (i) the score distribution over a candidate set and (ii) positive-vs-negative margins for hard negatives. We optionally add a weak reconstruction term only as a stabilizer.

Here we describe both the ranking-aligned training signals and how we optimize *discrete* document codes in a manner consistent with deployment. At indexing and inference, each token stores integer $\arg\max$ codes, but these hard decisions are non-differentiable. We therefore train with a differentiable surrogate based on soft assignments and progressively harden them to reduce train–test mismatch while keeping optimization stable. Training-cost details are reported in Appendix E.1.

### 5.1. Training Signals

The teacher induces a distribution over candidates by soft-maxing full-precision scores and the student does the same using compressed scores. With temperature $\tau$,

$$p_T(D \mid Q) = \frac{\exp(s(Q, D)/\tau)}{\sum_{D' \in \mathcal{C}(Q)} \exp(s(Q, D')/\tau)}, \quad (18)$$

$$p_S(D \mid Q) = \frac{\exp(\hat{s}(Q, D)/\tau)}{\sum_{D' \in \mathcal{C}(Q)} \exp(\hat{s}(Q, D')/\tau)}. \quad (19)$$

and we minimize

$$\mathcal{L}_{\text{list}}(Q) = \text{KL}(p_T(\cdot \mid Q) \,\|\, p_S(\cdot \mid Q)). \quad (20)$$

This loss aligns the student with the teacher's *relative* score structure inside $\mathcal{C}(Q)$, which is critical under late interaction because small changes to score gaps can flip ranks when candidates are close.

**Pairwise hard-negative loss.** While listwise matching shapes the overall score distribution, we also explicitly penalize violations where a hard negative outranks the positive:

$$\mathcal{L}_{\text{rank}}(Q) = \frac{1}{|\mathcal{N}(Q)|} \sum_{D^- \in \mathcal{N}(Q)} \log\Big(1 + \exp\big(\gamma\,[\hat{s}(Q, D^-)$$
$$-\hat{s}(Q, D^+)]\big)\Big), \quad (21)$$

where $\gamma$ controls sharpness: larger $\gamma$ concentrates gradient on near-ties and violations, which are the cases that most frequently change top-$k$ rankings under compression.

**Overall objective** A weak reconstruction term can stabilize optimization early in training (e.g., prevent assignment collapse):

$$\mathcal{L}_{\text{rec}}(D) = \frac{1}{n} \sum_{j=1}^{n} \|d_j - \hat{d}_j\|_2^2. \quad (22)$$

We optimize the weighted combination

$$\min_{\mathcal{E}, \mathcal{D}} \ \mathbb{E}_Q[\alpha\,\mathcal{L}_{\text{list}}(Q) + \beta\,\mathcal{L}_{\text{rank}}(Q)] + \lambda\,\mathbb{E}_D[\mathcal{L}_{\text{rec}}(D)]$$
$$\text{s.t.} \quad \text{Size}(\hat{\mathcal{D}}) \leq B. \quad (23)$$

In practice, we first choose $(M, K)$ to meet the target footprint and then train the compressor within that fixed budget.

### 5.2. Discrete Code Optimization

CrossQ produces per-stage assignment probabilities $\pi_{j,m,\cdot}$ (Eq. (14)) and decodes token embeddings using the expected codebook value (Eq. (15)). This yields a smooth relaxation of hard coding: when assignments are sharp, the forward pass closely matches the deployed representation, while gradients propagate through the assignment network and codebooks.

We sharpen assignments by annealing the quantizer temperature:

$$\tau_q(t) = \max\big(\tau_{\min},\,\tau_0\,r^t\big), \quad (24)$$

where $\tau_0$ is the initial temperature, $\tau_{\min}$ is a floor, $r \in (0, 1)$ is the decay rate and $t$ indexes training steps. As $\tau_q$ decreases, $\pi_{j,m,\cdot}$ approaches one-hot, so training-time decoding increasingly matches the hard $\arg\max$ codes used at indexing time. This is particularly important at small budgets, where small mismatches can flip near-tie orderings. At the most aggressive budget (2 B/token), we train with a straight-through estimator (STE): the forward pass uses hard codes $c_{j,m} = \arg\max_k \pi_{j,m,k}$, while the backward pass routes gradients through the soft probabilities $\pi_{j,m,\cdot}$. This makes the forward computation identical to deployment while retaining usable gradients. All 2 B/token results in Table 2 use STE.

To adapt the retriever to quantization noise without full retraining, we optionally perform *light fine-tuning*: we unfreeze the final projection head and the *last two* transformer blocks, keeping all earlier layers fixed. This modest adaptation improves ranking fidelity while preserving the overall ColBERT/ColBERTv2 training pipeline and keeping compute cost low.

## 6. Experimental Setup

We evaluate CrossQ in the regime that matters in practice: the index must be memory-resident and query-time latency must remain interactive. Accordingly, we compare methods under two constraints: (i) *matched nominal code budget* (2/4/8 B/token) for all *quantization* methods, with effective footprint including metadata reported separately and (ii) *comparable query-time latency* measured end-to-end under a fixed post-routing candidate budget. Unless noted otherwise, we compress only *document-side* token embeddings, query embeddings remain full precision, so quality changes are attributable to index compression rather than weaker query encoding. We report mean±std over three random

---

**Algorithm 1** CrossQ Training (Task-Aligned)

---

1: **Input:** training queries $\mathcal{Q}$; code design $(M, K)$, temperatures $\tau$ (listwise) and $\tau_q$ (quantizer)
2: Initialize codebooks $\{E^{(m)}\}$ and modules $(\phi, \rho, g)$, optionally set retriever parameters to unfreeze
3: **for** epoch = 1 to $T$ **do**
4:     Sample minibatch $(Q, D^+, \mathcal{N}(Q))$, set $\mathcal{C}(Q) = \{D^+\} \cup \mathcal{N}(Q)$
5:     Compute teacher scores $s(Q, D)$ for $D \in \mathcal{C}(Q)$ using full-precision document embeddings
6:     Encode each candidate $D$ with CrossQ at temperature $\tau_q$, compute compressed scores $\hat{s}(Q, D)$
7:     Compute $\mathcal{L}_{\text{list}}(Q)$ and $\mathcal{L}_{\text{rank}}(Q)$, optionally add $\lambda \mathcal{L}_{\text{rec}}$
8:     Update parameters with AdamW, anneal $\tau_q$ via Eq. (24)
9: **end for**
10: **Output:** trained CrossQ modules and codebooks, hard-code indexer uses $c_{j,m} = \arg\max_k \pi_{j,m,k}$

---

seeds. For each footprint, each quantization baseline is configured to meet the same budget constraint and then tuned for best retrieval quality.

All methods share the same retriever backbone, training data, tokenization/truncation and evaluation protocol. System baselines that tightly couple compression with routing (ColBERTv2/PLAID) are reported as references using their official default compression, for these methods we match the post-routing candidate pool size $|C(Q)|$ for latency comparisons, but do not retrain compression to meet our 2/4/8 B/token budgets.

For strictly footprint-matched quantization baselines, methods differ primarily in (i) the document-side representation stored in the index and (ii) the learning signal used to fit that representation. System references such as ColBERTv2/PLAID additionally include routing or pruning components and are therefore treated as candidate-matched references rather than strict footprint-matched baselines.

### 6.1. Datasets

We use MS MARCO passage ranking (Bajaj et al., 2018) as the in-domain benchmark. Following standard late-interaction evaluation (Khattab & Zaharia, 2020; Santhanam et al., 2022b), we report MRR@10 on the official development set. When space permits we also report Recall@$k$ (default $k \in \{10, 100, 1000\}$), since late-interaction models are commonly used as high-recall first-stage retrievers.

To assess robustness under domain shift, we evaluate on BEIR (Thakur et al., 2021) and report average nDCG@10 across datasets. Because cross-domain behavior can vary substantially by dataset, we include a full per-dataset breakdown in the appendix and summarize representative datasets in Table 3.

### 6.2. Retriever Backbone and Indexing Protocol

**Backbone and encoding.** We adopt a ColBERT-style late-interaction retriever (Khattab & Zaharia, 2020) and follow ColBERTv2 training/tokenization conventions where applicable (Santhanam et al., 2022b). At indexing time, each passage is encoded into token embeddings $D = \{d_j\}_{j=1}^n$ (after truncation to $n_{\max}$), which are then converted to discrete codes and stored in a flattened token-code index with per-document offsets and lengths.

**CrossQ indexing-time context (not stored).** CrossQ additionally computes a lightweight, query-independent document context vector $h(D)$ during indexing to condition code selection. This affects *which codes are chosen* but does not alter the inference-time scoring operator: scoring remains max-sim late interaction as in Eq. 4. Crucially, $h(D)$ **is not stored in the index**, only integer token codes are stored. Operationally, CrossQ trades modest extra indexing-time compute for better codes while keeping query-time scoring as standard lookup-table evaluation for additive codebooks.

### 6.3. Baselines and CrossQ Variants

We compare to baselines that isolate three factors: (i) token-wise quantization quality under the same bytes/token, (ii) system-level accelerations for late interaction and (iii) CrossQ design choices.

**Token-wise quantization.** We compare against PQ/OPQ applied independently per token (Jégou et al., 2011; Ge et al., 2013), residual quantization (RQ), and a token-wise conditional residual quantizer (QINCo2-style). We also include a token-wise conditional variant trained with our ranking-aligned losses to isolate the effect of *conditionality* without cross-token context. All token-wise baselines quantize each token independently and do not use document-level context.[1]

**Late-interaction acceleration systems.** We compare to ColBERTv2 compression/distillation (Santhanam et al., 2022b) and PLAID routing/pruning with optimized late-interaction execution (Santhanam et al., 2022a). ColBERTv2's default compression yields $\sim 8$ B/token and is not retrained for lower budgets. For PLAID, we tune routing thresholds to match the target candidate count $|C(Q)| = 20{,}000$. Since these systems use official default compression, they serve as candidate-matched system references rather than strictly footprint-matched baselines at 2/4/8 B/token.

To attribute gains, we evaluate: (a) **CrossQ (recon-only)** trained with reconstruction only. (b) **CrossQ (no-ctx)** removing document context $h(D)$ (token-local conditioning only). (c) **CrossQ (no-list)** removing listwise score distil-

---

[1]QINCo2 targets single-vector ANN. We apply it independently to each document token embedding while keeping late-interaction scoring unchanged.

lation. (d) **CrossQ + light FT**, which unfreezes the last 2 retriever blocks plus the projection head.

Light fine-tuning (unfreezing the last 2 transformer blocks + projection head) was applied only to CrossQ in our main experiments. We include an additional ablation in Appendix G showing that applying identical fine-tuning to token-wise baselines yields smaller gains (+0.002 MRR@10 for OPQ at 4 B/token vs. +0.006 for CrossQ), confirming our improvements stem primarily from the quantization architecture rather than adaptation capacity.

### 6.4. Budgets, Metrics and Measurement Protocol

We compare quantization methods at matched nominal code budgets: `Small`=2, `Medium`=4 and `Large`=8 bytes/token. We instantiate these budgets with additive codebooks using $K$=256 and $M \in \{2, 4, 8\}$ (16/32/64 bits). Per-document metadata such as offsets, lengths and alignment is accounted for separately and reported as effective footprint in Appendix D. Each quantization baseline is configured to use the same nominal code budget and then tuned for best retrieval quality.

On MS MARCO we report MRR@10 (primary) and Recall@$k$ (default $k$=10 optionally $k$=100, 1000). On BEIR we report average nDCG@10 and include representative per-dataset results in Table 3 (full breakdown in the appendix).

Because our target is ranking behavior under max-sim, not reconstruction error, we report two diagnostics: (i) *teacher–student margin error* $\Delta_S - \Delta_T$ over hard negatives, summarized by its distribution (e.g., Fig. 2) and (ii) the *listwise KL* between teacher and student candidate distributions (Eq. 20). The **teacher** uses full-precision document embeddings (no compression).

**Efficiency Measurement** We report (i) index footprint (bytes/token and total GB), (ii) indexing throughput (docs/sec) and (iii) end-to-end query-time latency (p50/p95) and throughput (QPS). Latency is measured with batch size $B$=1 unless stated otherwise.

We fix the post-routing candidate pool to $|C(Q)|$=20,000 and use identical query encoding and scoring across methods. We report p50/p95 latency over 1,000 deterministic dev queries on 4×A100 80GB (scoring) and a 32-core CPU (index ops), with warm-cache runs and explicit GPU synchronization. See Appendix E for full details.

## 7. Results

We report retrieval quality at matched nominal code budgets (2/4/8 bytes per token) and compare against strong baselines under the same evaluation protocol. Numbers are mean±std over three random seeds (Section 6).

**CrossQ improves the quality–footprint trade-off.** Table 2 shows consistent gains across all budgets. Improvements are most pronounced at 2 B/token, where max-sim

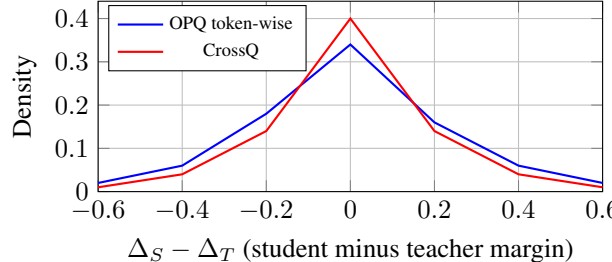

*Figure 2.* **Margin error distribution.** CrossQ concentrates mass near zero, indicating better teacher-margin preservation and fewer near-tie flips.

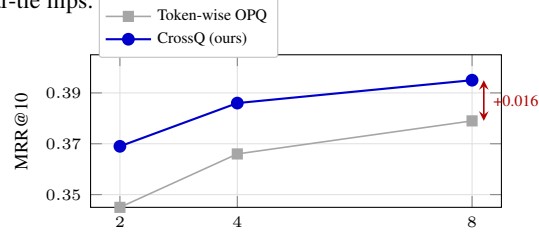

*Figure 3.* **Scaling with budget via $M$.** Increasing $M$ from 2 to 8 (at $K = 256$) raises the budget from 2 to 8 B/token. CrossQ converts additional capacity into retrieval quality more effectively than token-wise OPQ.

late interaction is highly sensitive to quantization noise. At 2 B/token, CrossQ improves over the strongest strictly footprint-matched quantization baseline (Token-wise Conditional RQ, rank) from 0.359 to **0.369** MRR@10. It also improves over the candidate-matched PLAID system reference from 0.357 to **0.369** MRR@10 and from 0.812 to **0.826** Recall@10. Gains persist at higher budgets, with CrossQ reaching **0.386** MRR@10 at 4 B/token and **0.395** at 8 B/token, shifting the curve rather than optimizing a single point. As we increase the number of additive stages $M$ (and thus the bytes/token budget at fixed $K$), CrossQ improves more than token-wise OPQ (Figure 3). This indicates conditional assignment more effectively allocates additional capacity toward ranking preservation, rather than uniformly reducing reconstruction error.

The full-precision ColBERT teacher obtains 0.408 MRR@10 on MS MARCO dev under the same evaluation protocol; therefore, CrossQ + light FT at 8 B/token retains approximately 98% of teacher MRR@10.

**Light fine-tuning improves ranking fidelity with minimal adaptation.** Unfreezing only the projection head and the final two encoder blocks yields further gains without full retraining. At 2 B/token, **CrossQ + light FT** reaches **0.374** MRR@10 (+0.005) and at 4 B/token improves MRR@10 from 0.386 to **0.392**. This suggests modest adaptation can absorb residual quantization error and recover teacher-like score ordering.

**CrossQ improves robustness under domain shift.** On the BEIR subset reported in Appendix G.1, CrossQ improves average nDCG@10 at 4 B/token from 0.387 to **0.396** over the strongest candidate-matched system reference and

*Table 2.* **Quality vs. footprint.** Higher is better. Mean±std over 3 seeds. PQ/OPQ/RQ/token-wise conditional rows are strictly footprint-matched at the specified nominal code budgets. PLAID is included as a candidate-matched system reference under the same post-routing candidate budget, but its official compression/routing stack is not retrained to satisfy each 2/4/8 B/token budget. **Token-wise Conditional (RQ)** denotes QINCo2-style residual conditional quantization without document context; "rank" uses our listwise+pairwise losses.

| Method | 2 B/token | | 4 B/token | | 8 B/token | |
|---|---|---|---|---|---|---|
| | **MRR@10** | **R@10** | **MRR@10** | **R@10** | **MRR@10** | **R@10** |
| Product Quantization (PQ) | $0.341_{\pm0.002}$ | $0.792_{\pm0.003}$ | $0.360_{\pm0.002}$ | $0.818_{\pm0.003}$ | $0.375_{\pm0.002}$ | $0.832_{\pm0.003}$ |
| Optimized Product Quantization (OPQ) | $0.345_{\pm0.002}$ | $0.796_{\pm0.003}$ | $0.366_{\pm0.002}$ | $0.823_{\pm0.003}$ | $0.379_{\pm0.002}$ | $0.836_{\pm0.003}$ |
| Token-wise Conditional (RQ, recon) | $0.352_{\pm0.002}$ | $0.804_{\pm0.003}$ | $0.372_{\pm0.002}$ | $0.828_{\pm0.003}$ | $0.384_{\pm0.002}$ | $0.840_{\pm0.003}$ |
| Token-wise Conditional (RQ, rank) | $0.359_{\pm0.002}$ | $0.811_{\pm0.003}$ | $0.371_{\pm0.002}$ | $0.830_{\pm0.003}$ | $0.387_{\pm0.002}$ | $0.842_{\pm0.003}$ |
| PLAID (cand.-matched ref.) | $0.357_{\pm0.002}$ | $0.812_{\pm0.003}$ | $0.381_{\pm0.002}$ | $0.832_{\pm0.003}$ | $0.389_{\pm0.002}$ | $0.840_{\pm0.003}$ |
| **CrossQ** | $\mathbf{0.369}_{\pm0.001}$ | $\mathbf{0.826}_{\pm0.002}$ | $\mathbf{0.386}_{\pm0.001}$ | $\mathbf{0.842}_{\pm0.002}$ | $\mathbf{0.395}_{\pm0.001}$ | $\mathbf{0.850}_{\pm0.002}$ |
| **CrossQ + Fine-Tuning** | $\mathbf{0.374}_{\pm0.001}$ | $\mathbf{0.831}_{\pm0.002}$ | $\mathbf{0.392}_{\pm0.001}$ | $\mathbf{0.848}_{\pm0.002}$ | $\mathbf{0.399}_{\pm0.001}$ | $\mathbf{0.854}_{\pm0.002}$ |

*Table 3.* **Representative nDCG@10 breakdown (4 B/token).** nDCG@10 per dataset. Mean±std over 3 seeds. Full BEIR-subset averages are reported in Appendix G.1.

| Method | MSMARCO | TREC-COVID | NFCorpus | HotpotQA | FiQA | ArguAna | Touché-2020 |
|---|---|---|---|---|---|---|---|
| OPQ (token-wise) | $0.395_{\pm0.005}$ | $0.482_{\pm0.008}$ | $0.315_{\pm0.009}$ | $0.583_{\pm0.007}$ | $0.264_{\pm0.005}$ | $0.351_{\pm0.012}$ | $0.248_{\pm0.009}$ |
| Token-wise conditional | $0.408_{\pm0.004}$ | $0.491_{\pm0.007}$ | $0.328_{\pm0.008}$ | $0.590_{\pm0.006}$ | $0.272_{\pm0.004}$ | $0.364_{\pm0.010}$ | $0.253_{\pm0.008}$ |
| PLAID | $0.420_{\pm0.004}$ | $0.503_{\pm0.006}$ | $0.339_{\pm0.007}$ | $0.601_{\pm0.005}$ | $0.281_{\pm0.003}$ | $0.378_{\pm0.009}$ | $0.260_{\pm0.007}$ |
| **CrossQ** | $\mathbf{0.435}_{\pm0.003}$ | $\mathbf{0.510}_{\pm0.005}$ | $\mathbf{0.347}_{\pm0.006}$ | $\mathbf{0.612}_{\pm0.004}$ | $\mathbf{0.289}_{\pm0.003}$ | $\mathbf{0.392}_{\pm0.008}$ | $\mathbf{0.268}_{\pm0.006}$ |
| **CrossQ + light FT** | $\mathbf{0.443}_{\pm0.003}$ | $\mathbf{0.518}_{\pm0.004}$ | $\mathbf{0.354}_{\pm0.005}$ | $\mathbf{0.620}_{\pm0.003}$ | $\mathbf{0.295}_{\pm0.002}$ | $\mathbf{0.401}_{\pm0.007}$ | $\mathbf{0.274}_{\pm0.005}$ |

*Table 4.* Reconstruction is a weak proxy. Correlation between reconstruction fidelity (MSE) and MS MARCO MRR@10 across methods and budgets.

| Setting | Pearson $r$ | Spearman $\rho$ |
|---|---|---|
| Token-wise baselines (varied $M$, $K$) | 0.41 | 0.36 |
| CrossQ variants (ctx/no-ctx losses) | 0.28 | 0.22 |
| All points (mixed methods) | 0.33 | 0.30 |

**CrossQ + light FT** reaches **0.403**. On the representative datasets in Table 3, CrossQ consistently improves over OPQ, token-wise conditional quantization and PLAID, suggesting ranking-aligned compression transfers better beyond the training distribution.

**Why CrossQ helps: ranking preservation matters more than reconstruction.** Token reconstruction error correlates weakly with retrieval quality (Table 4), reflecting the mismatch between MSE-style objectives and max-sim scoring. In contrast, CrossQ better preserves teacher score margins: the distribution of $\Delta_S - \Delta_T$ concentrates closer to zero than OPQ (Figure 2), reducing near-tie inversions. This supports the view that preserving relative score structure is key under late interaction.

CrossQ conditions code selection on a lightweight document context $h(D)$ computed at indexing time. Appendix G.4 further shows that CrossQ concentrates lower error on high-winner-frequency tokens (Table 24) and reduces winner-identity flips relative to OPQ (Table 25), explaining why context improves ranking preservation.

**Latency vs. footprint.** Full-precision ColBERT remains the fastest configuration when the index fits entirely in fast memory (Table 15), benefiting from contiguous memory access and optimized dense GPU kernels. Code-based representations introduce indirection (codebook lookup, gather-heavy access) and incur a modest scoring overhead. CrossQ therefore shifts the footprint–quality frontier rather than the latency–quality frontier: it enables memory-feasible serving in regimes where full-precision late-interaction indices exceed available RAM, where paging and IO/cache pressure would otherwise dominate end-to-end performance. We report scoring-only and end-to-end timings separately in Appendix E.

## 8. Ablation Studies

We ablate CrossQ's two core ingredients at a fixed footprint (4 B/token): (i) the *cross-token document context* $h(D)$ for conditioning code selection and (ii) the *task-aligned training signals* (listwise score distillation and pairwise hard-negative supervision). All ablations share the same backbone retriever, code design $(M, K)$, candidate construction and optimization schedule we change only the component under test. The goal is to attribute gains to *ranking preservation under max-sim* rather than lower average reconstruction error.

**Context provides the largest single contribution.** Removing context ("no-ctx") yields the largest drop on MS MARCO at 4 B/token (MRR@10: $0.386 \rightarrow 0.371$,

*Table 5.* Ablation (4 B/token, MS MARCO). All variants share the additive quantizer architecture. "Token-local" = code selection conditioned only on $d_j$ (no document context).

| Variant | MRR@10 |
|---|---|
| CrossQ (full: context + listwise + pairwise) | **0.386** |
| − context $h(D)$ | 0.371 (−0.015) |
| − listwise KL | 0.377 (−0.009) |
| − context & listwise | 0.364 (−0.022) |
| − context & ranking (recon only) | 0.358 (−0.028) |

−0.015), consistent with $h(D)$ enabling within-document capacity allocation toward tokens that are more likely to become max-sim winners.

**Ranking-aligned losses complement reconstruction.** Removing listwise KL ("no-list") reduces MRR@10 (0.386 → 0.377, −0.009) and hurts BEIR, suggesting that matching *relative score gaps* within candidate sets improves robustness beyond reconstruction-style training. Training with reconstruction alone ("rec-only") performs worst at the same budget (MRR@10: 0.358, −0.028 vs. full CrossQ), while removing both context and listwise supervision remains below full CrossQ (MRR@10: 0.364, −0.022), indicating the signals address complementary failure modes.

**Efficiency.** CrossQ adds indexing-time compute beyond PQ due to context extraction and conditional code assignment, reducing indexing throughput from 2,400 docs/sec (PQ) to 1,850 docs/sec (CrossQ, ∼1.3× slower). Query-time scoring remains max-sim with integer-coded table lookups under additive decoding (Eq. 17). Unless stated otherwise, the p50 latency reported here refers to the scoring stage under a fixed post-routing candidate pool $|C(Q)|$=20,000 (3.6 ms for CrossQ vs. 3.2 ms for PQ). End-to-end latency, including candidate construction and scoring, is reported in Appendix E.
We use $\alpha$=1.0, $\beta$=1.0, $\lambda$=0.01 and $\tau$=0.05. Sweep ranges and selected configurations are reported in Appendix F.

**Disentangling capacity from cross-token information.** The context pathway adds ∼131K parameters over the no-ctx variant, raising the question of whether the +0.015 MRR@10 gap reflects information or capacity. We run two controls at 4 B/token (Table 6): (i) a *param-matched no-ctx* variant that scales the token-local MLP to exactly match CrossQ's 1.31M parameter count and (ii) a *shuffled-context* variant with the full context pathway intact (identical params and compute) but fed token sets shuffled across documents, destroying per-document signal. Extra capacity alone recovers only +0.001 of the +0.015 gap. The shuffled-context variant, despite matching CrossQ's architecture exactly, drops to 0.370 — slightly below no-ctx — indicating decorrelated context injects noise rather than serving as a neutral capacity baseline. Table 11 corroborates: doubling $d_c$ from 256 to 512 adds another 131K parameters with zero MRR gain.

*Table 6.* Controls disentangling capacity from cross-token information (4 B/token, MS MARCO).

| Variant | Params | MRR@10 |
|---|---|---|
| CrossQ (full: ctx + listwise + pairwise) | 1.31M | **0.386** |
| CrossQ (no-ctx) | 1.18M | 0.371 |
| no-ctx (param-matched) | 1.31M | 0.372 |
| ctx (shuffled $h$) | 1.31M | 0.370 |

## 9. Conclusion and Limitations

**CrossQ** is a cross-token conditional quantizer for late-interaction retrieval targeting ranking preservation over token reconstruction. By conditioning code selection on document context and training with score-level supervision, it improves MS MARCO effectiveness, BEIR robustness and reduces near-tie inversions. It adds ∼1.3× offline indexing cost (amortized over queries) but yields much smaller indices and enables memory-feasible late-interaction serving when full-precision indices exceed available memory. Limitations include potential max-sim flips at extreme compression and tailoring to ColBERT-style scoring; other operators may require adapted training signals.

## Impact Statement

This paper advances efficient neural information retrieval by reducing the storage footprint of late-interaction indices while preserving ranking quality. CrossQ achieves up to 64× raw token-storage reduction, approximately 61× compression when metadata is included and approximately 58× compression under conservative padding/alignment accounting. This can lower the storage and deployment barriers for high-quality retrieval systems and may reduce the energy costs associated with large-scale retrieval infrastructure. CrossQ does not address the broader social risks of retrieval systems. It inherits the biases, omissions and failure modes of the underlying retriever and corpus. Making strong retrieval systems cheaper to deploy may also widen their use in harmful settings, including biased ranking or surveillance-oriented applications. We do not introduce new training data or new ranking objectives intended to mitigate these risks, therefore, these concerns remain comparable to those of standard ColBERT-style late-interaction retrievers.

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

# A. Appendix Overview

This appendix provides comprehensive supplementary material for CrossQ, organized as follows:

- **Appendix B**: Complete implementation and training details, including architecture specifications, loss derivations and optimization procedures.
- **Appendix C**: Detailed baseline descriptions and implementation notes for fair comparison.
- **Appendix D**: Rigorous index footprint accounting with worked examples.
- **Appendix E**: Efficiency measurement protocol ensuring reproducible latency comparisons.
- **Appendix F**: Hyperparameter sweep procedures and sensitivity analysis.
- **Appendix G**: Extended results, ablations, diagnostics and failure analysis.
- **Appendix H**: Mathematical derivations and proofs.
- **Appendix I**: Computational complexity analysis.
- **Appendix J**: Additional experiments addressing indexing time, negative quality, longer documents and memory overhead.
- **Appendix K**: Extended discussion of limitations regarding generalization, scalability, related work, failure cases, STE stability and hyperparameter sensitivity.

We follow the ColBERT/ColBERTv2 pipeline for query and document encoding and late-interaction scoring, modifying only the *document-side* representation via quantization.

# B. Implementation and Training Details

## B.1. Retriever Backbone and Tokenization

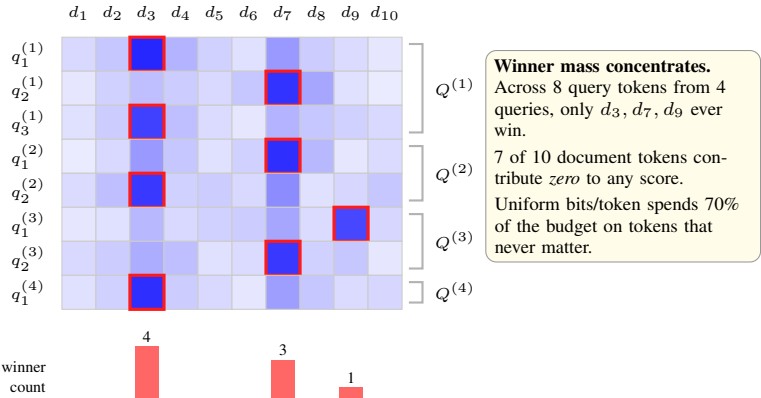

*Figure 4.* **Why uniform precision wastes budget: max-sim winners are sparse.** Heatmap shows similarity scores $\langle q_i, d_j \rangle$ between query tokens (rows, grouped by query) and document tokens (columns) for one document. Darker = higher similarity. Red boxes mark per-row argmax — the "winner" token contributing to the score. The histogram below counts winner frequency per document token. Only a few columns ($d_3, d_7, d_9$) ever win, while $d_1, d_2, d_4, d_5, d_6, d_8, d_{10}$ never appear in any score. CrossQ's document context $h(D)$ provides a query-independent signal that can bias code selection toward tokens statistically more likely to be high-leverage, enabling within-document precision reallocation without storing the context.

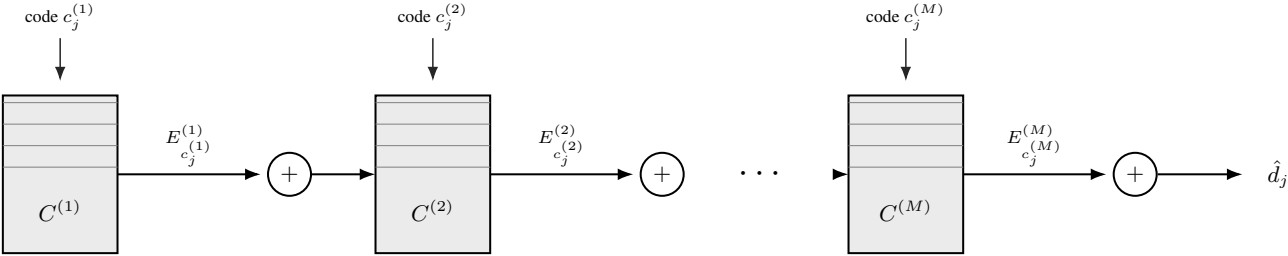

*Figure 5.* **CrossQ decoding.** Stored codes $(c_j^{(1)}, \ldots, c_j^{(M)})$ index into $M$ shared codebooks $C^{(m)}$ to retrieve codewords $E_{c_j^{(m)}}^{(m)}$, which are summed to reconstruct the token embedding: $\hat{d}_j = \sum_{m=1}^{M} E_{c_j^{(m)}}^{(m)}$. Unlike residual quantizers (QINCo, RQ), CrossQ uses a *parallel additive* structure with no sequential dependency between stages, enabling per-query lookup-table scoring (Fig. 7).

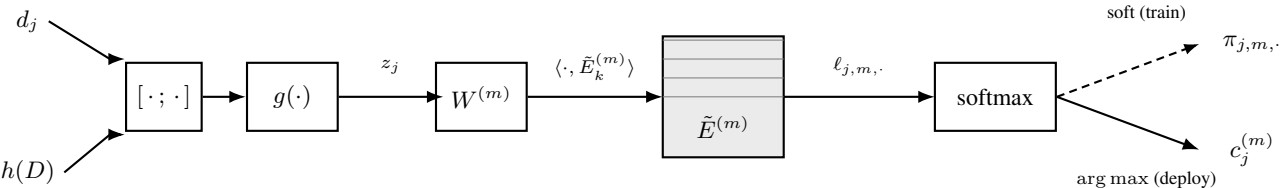

*Figure 6.* **Conditional code selector at stage** $m$**.** The token embedding $d_j$ and document context $h(D)$ are concatenated and passed through a small MLP $g(\cdot)$ to produce a conditional representation $z_j$. A learned projection $W^{(m)}$ and assignment keys $\tilde{E}^{(m)}$ produce logits $\ell_{j,m,k} = \langle W^{(m)} z_j, \tilde{E}_k^{(m)} \rangle$ over the $K$ codebook entries. At training time, temperature-controlled softmax yields differentiable assignments $\pi_{j,m,\cdot}$ (Eq. 14) at indexing time, the $\arg\max$ selects the stored code $c_j^{(m)}$. Document context $h(D)$ is computed once per document at indexing time and discarded; only the integer code $c_j^{(m)}$ is stored.

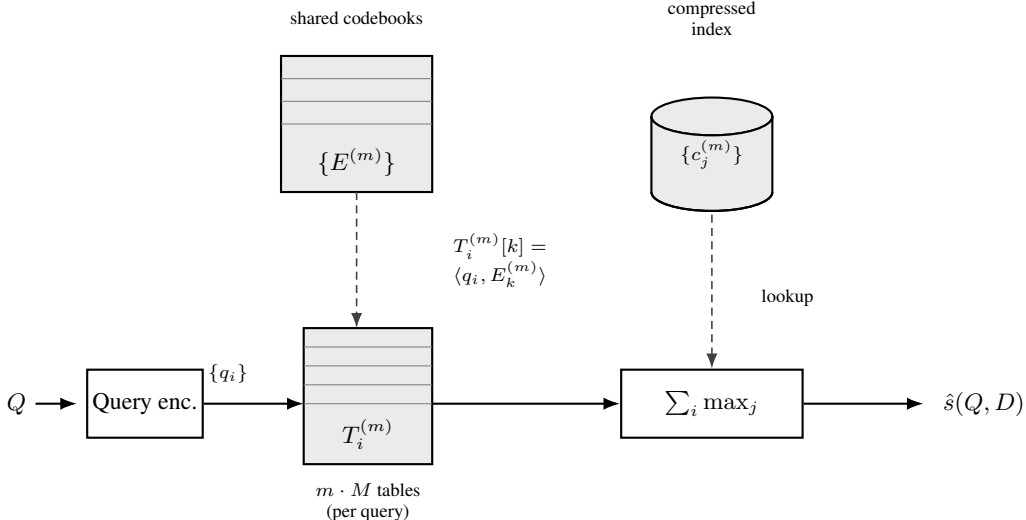

*Figure 7.* **Query-time scoring with lookup tables.** Given query embeddings $\{q_i\}$, we precompute $mM$ lookup tables $T_i^{(m)}[k] = \langle q_i, E_k^{(m)} \rangle$ — one per query token, per codebook — by dotting query tokens against the shared codebooks. This is a one-time cost per query (independent of corpus size). Document scoring then reduces to $mM$ integer lookups per document followed by max-sim aggregation: $\hat{s}(Q, D) = \sum_i \max_j \sum_m T_i^{(m)}[c_j^{(m)}]$. No $d$-dimensional floating-point dot products against stored document vectors occur in the hot path; the index stores only integer codes.

**Base retriever architecture.** We use a ColBERT-style late-interaction retriever (Khattab & Zaharia, 2020) with Col-BERTv2 training conventions (Santhanam et al., 2022b). The retriever consists of:

- **Encoder**: BERT-base (110M parameters) with a linear projection head mapping from hidden dimension $d_{\text{hidden}} = 768$ to embedding dimension $d = 128$.
- **Query encoder**: Prepends [Q] marker token applies learned query augmentation.
- **Document encoder**: Prepends [D] marker token no augmentation.
- **Scoring**: Max-sim late interaction (Eq. 4).

The retriever maps each query $Q$ and document $D$ into token-level embeddings in $\mathbb{R}^d$ and scores pairs using:

$$s(Q, D) = \sum_{i=1}^{m} \max_{j \in [n]} \langle q_i, d_j \rangle, \tag{25}$$

where $q_i \in \mathbb{R}^d$ and $d_j \in \mathbb{R}^d$ are $\ell_2$-normalized token embeddings.

**Tokenization and sequence lengths.** We use the WordPiece tokenizer from the underlying BERT encoder with vocabulary size 30,522. Sequence length constraints are:
All methods use identical tokenization and truncation performance differences arise solely from document-side compression.
For documents shorter than $n_{\max}$, we do not pad the actual token count $n \leq n_{\max}$ is stored in metadata.

*Table 7.* Tokenization parameters.

| Parameter | Symbol | Value |
|---|---|---|
| Query max length | $m_{\max}$ | 32 |
| Document max length | $n_{\max}$ | 180 |
| Query augmentation tokens | — | 8 (masked) |
| Special tokens per sequence | — | 2 (`[CLS]`, `[SEP]`) |

**Normalization.** We $\ell_2$ normalize token embeddings before similarity computation:

$$\tilde{q}_i = \frac{q_i}{\|q_i\|_2}, \qquad \tilde{d}_j = \frac{d_j}{\|d_j\|_2}. \tag{26}$$

This ensures inner products correspond to cosine similarity: $\langle \tilde{q}_i, \tilde{d}_j \rangle = \cos(\theta_{ij})$. If the backbone already applies normalization (as in ColBERTv2), we do not apply a second normalization.

### B.2. CrossQ Quantizer Architecture

CrossQ compresses only document token embeddings $\{d_j\}_{j=1}^n$ into integer codes and reconstructs $\hat{d}_j$ via additive codebooks. Figure 8 illustrates the complete architecture.

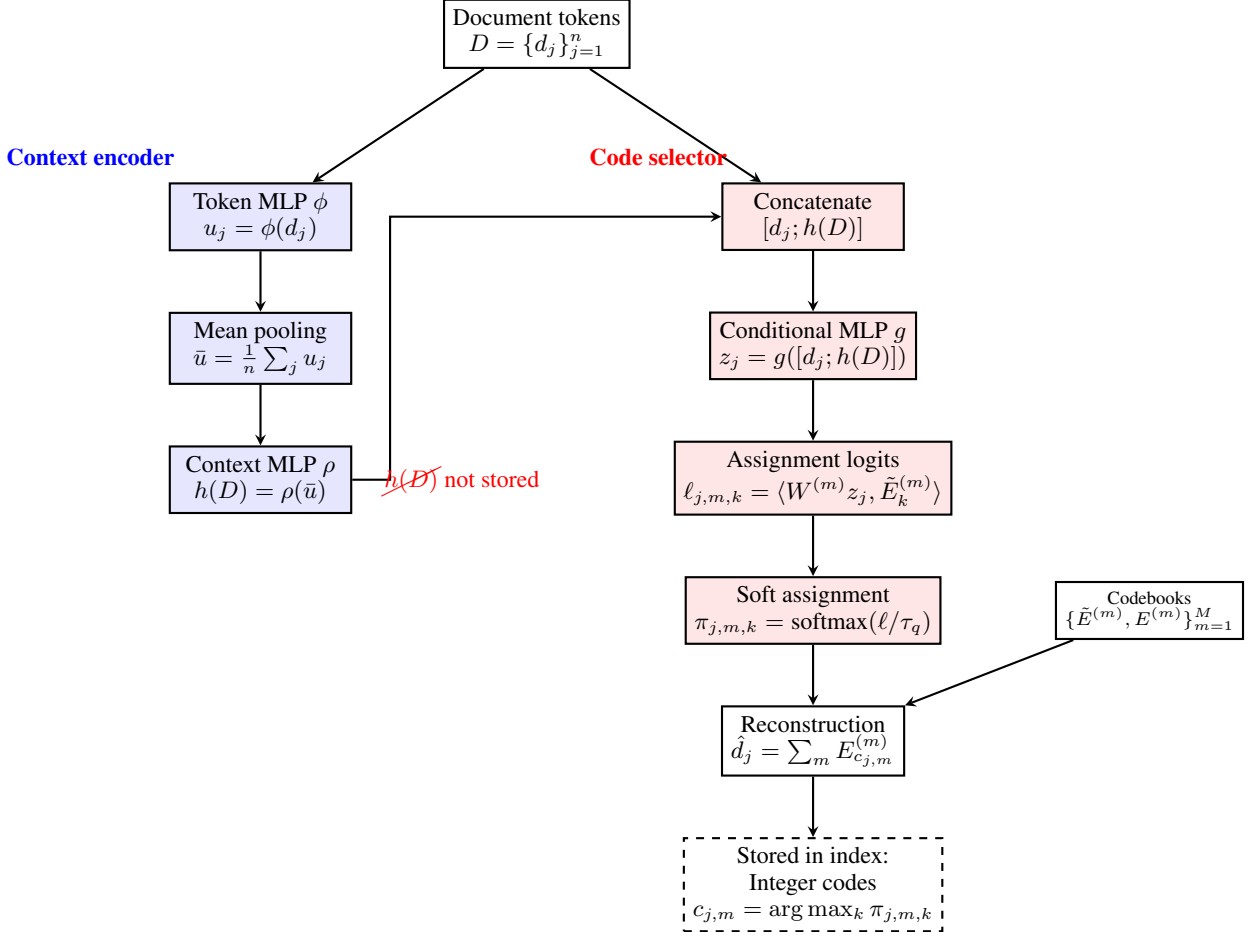

*Figure 8.* **CrossQ architecture.** The document context $h(D)$ is computed via a permutation-invariant set encoder (blue branch) and conditions the code selection network (red branch). At indexing time, only hard codes $c_{j,m} = \arg\max_k \pi_{j,m,k}$ are stored $h(D)$ is discarded and not stored in the index.

**Codebooks and reconstruction.** We use $M$ additive codebooks $\{E^{(m)}\}_{m=1}^M$ with $K$ entries each, where $E_k^{(m)} \in \mathbb{R}^d$. Each token is encoded as $M$ integers $\{c_{j,m}\}_{m=1}^M$ with $c_{j,m} \in [K]$ and reconstructed by:

$$\hat{d}_j = \sum_{m=1}^M E_{c_{j,m}}^{(m)}. \tag{27}$$

The additive structure has two key advantages for late interaction:

1. **Decomposable scoring**: Query-document similarity decomposes into $M$ lookup tables (Section B.5).

2. **Incremental refinement**: Each codebook stage refines the reconstruction, analogous to residual quantization but with parallel (not sequential) structure.

**Assignment keys vs. reconstruction values.** We maintain separate assignment keys $\tilde{E}^{(m)}$ and reconstruction values $E^{(m)}$:

• **Assignment keys** $\tilde{E}^{(m)} \in \mathbb{R}^{K \times d_z}$: Used to compute assignment logits. These parameterize the "assignment geometry" which tokens map to which codes.

• **Reconstruction values** $E^{(m)} \in \mathbb{R}^{K \times d}$: The actual codebook entries stored and used at inference for scoring.

When tied ($\tilde{E}^{(m)} \equiv E^{(m)}$), the model has fewer parameters but less flexibility. We find untied keys improve performance by 0.003 MRR@10 at 4 B/token (Table 8).

*Table 8.* Effect of tied vs. untied assignment keys (4 B/token, MS MARCO).

| Configuration | MRR@10 | Parameters |
|---|---|---|
| Tied ($\tilde{E}^{(m)} \equiv E^{(m)}$) | 0.383 | 1.05M |
| Untied | **0.386** | 1.31M |

**Initialization.** We explore three initialization strategies:

1. **Random Gaussian**: $E_k^{(m)} \sim \mathcal{N}(0, 0.01 \cdot I_d)$

2. **K-means**: Run K-means on a sample of document tokens, use centroids.

3. **OPQ warm-start**: Initialize from pre-trained OPQ codebooks.

Table 9 shows initialization has modest impact we use random Gaussian for simplicity.

*Table 9.* Effect of codebook initialization (4 B/token, MS MARCO).

| Initialization | MRR@10 | Training time |
|---|---|---|
| Random Gaussian | 0.386 | 1.0× |
| K-means | 0.387 | 1.1× |
| OPQ warm-start | 0.388 | 0.9× |

### B.3. Document Context Encoder

CrossQ conditions code selection on a query-independent document context $h(D)$ computed at indexing time. Crucially, $h(D)$ *is not stored in the index* only integer token codes are stored.

**DeepSets-style set encoder.** We use a permutation-invariant encoder following DeepSets (Zaheer et al., 2017):

$$u_j = \phi(d_j) \in \mathbb{R}^{d_c}, \tag{28}$$

$$h(D) = \rho\left(\frac{1}{n}\sum_{j=1}^n u_j\right) \in \mathbb{R}^{d_c}, \tag{29}$$

where $\phi : \mathbb{R}^d \to \mathbb{R}^{d_c}$ and $\rho : \mathbb{R}^{d_c} \to \mathbb{R}^{d_c}$ are small MLPs.

*Table 10.* Comparison of context computation methods (4 B/token).

| Method | MRR@10 | Compute (ms/doc) | Permutation invariant? |
|---|---|---|---|
| Mean pooling (no MLP) | 0.378 | 0.02 | Yes |
| CLS token | 0.375 | 0.01 | No |
| Attention pooling | 0.384 | 0.15 | Yes |
| DeepSets (ours) | **0.386** | 0.08 | Yes |
| Transformer (1 layer) | 0.387 | 0.42 | No |

**Why DeepSets?** We considered several alternatives for computing document context:

DeepSets offers the best trade-off: it outperforms simple mean pooling (+0.008 MRR@10) while remaining permutation-invariant and computationally efficient. The Transformer variant provides marginal gains (+0.001) but at $5\times$ higher compute cost and loses permutation invariance.

Attention pooling remains competitive but is slightly below DeepSets on the main 4 B/token setting (0.384 vs. 0.386 MRR@10) while nearly doubling compute overhead (0.15 vs. 0.08 ms/doc). We therefore prioritize the efficiency and strict permutation invariance of the DeepSets approach for the general case.

**MLP parameterization.** Both $\phi$ and $\rho$ are two-layer MLPs with identical structure:

$$\mathrm{MLP}(x) = W_2 \cdot \sigma(\mathrm{LN}(W_1 x + b_1)) + b_2, \tag{30}$$

where:

- $\sigma(\cdot)$ is GELU activation (Hendrycks & Gimpel, 2016)
- $\mathrm{LN}(\cdot)$ is LayerNorm applied after the hidden layer
- Hidden dimension is $2 \times d_c$ (expansion factor 2)
- Dropout (rate 0.1) is applied after $\sigma(\cdot)$

**Context dimension.** The context dimension $d_c$ controls the capacity of the shared document signal. We ablate $d_c$ in Table 11.

*Table 11.* Effect of context dimension $d_c$ (4 B/token, MS MARCO).

| $d_c$ | MRR@10 | Parameters | Compute overhead |
|---|---|---|---|
| 64 | 0.381 | +33K | +3% |
| 128 | 0.384 | +66K | +5% |
| 256 | **0.386** | +131K | +8% |
| 512 | 0.386 | +262K | +12% |

Performance saturates at $d_c = 256$; we use this value in all experiments.

### B.4. Conditional Code Selection Network

**Conditional token representation.** Given token embedding $d_j \in \mathbb{R}^d$ and document context $h(D) \in \mathbb{R}^{d_c}$, we form a conditional representation:

$$z_j = g([d_j; h(D)]) \in \mathbb{R}^{d_z}, \tag{31}$$

where $[\cdot; \cdot]$ denotes concatenation and $g : \mathbb{R}^{d+d_c} \to \mathbb{R}^{d_z}$ is a two-layer MLP with GELU activation. We use $d_z = 128$ in all experiments.

The concatenation allows $g$ to learn interactions between token-local features ($d_j$) and document-global features ($h(D)$). This is the key mechanism enabling cross-token capacity allocation.

**Per-codebook logits.** For each codebook $m \in [M]$, we compute logits over all $K$ entries:

$$\ell_{j,m,k} = \left\langle W^{(m)} z_j, \tilde{E}_k^{(m)} \right\rangle, \quad k \in [K], \tag{32}$$

where $W^{(m)} \in \mathbb{R}^{d_z \times d_z}$ is a learned projection. This inner-product formulation is efficient: we can precompute $W^{(m)} z_j$ once per token and then compute $K$ dot products against assignment keys.

**Soft assignments and temperature.** We convert logits to assignment probabilities via temperature-controlled softmax:

$$\pi_{j,m,k} = \frac{\exp(\ell_{j,m,k}/\tau_q)}{\sum_{k'=1}^{K} \exp(\ell_{j,m,k'}/\tau_q)}. \tag{33}$$

The temperature $\tau_q$ controls assignment sharpness:
- High $\tau_q$ (e.g., 1.0): Soft, diffuse assignments enabling gradient flow.
- Low $\tau_q$ (e.g., 0.05): Sharp, nearly one-hot assignments matching deployment.

During training, we use soft reconstruction:

$$\hat{d}_j = \sum_{m=1}^{M} \sum_{k=1}^{K} \pi_{j,m,k} \cdot E_k^{(m)}. \tag{34}$$

At indexing time, we store hard codes $c_{j,m} = \arg\max_k \pi_{j,m,k}$ for each token and stage.

### B.5. Scoring Decomposition for Efficient Inference

Because CrossQ stores additive codes, token similarities decompose into lookup table operations. This section derives the efficient scoring procedure.

**Similarity decomposition.** For a query token $q_i$ and compressed document token $\hat{d}_j$:

$$\langle q_i, \hat{d}_j \rangle = \left\langle q_i, \sum_{m=1}^{M} E_{c_{j,m}}^{(m)} \right\rangle \tag{35}$$

$$= \sum_{m=1}^{M} \left\langle q_i, E_{c_{j,m}}^{(m)} \right\rangle \tag{36}$$

$$= \sum_{m=1}^{M} T_i^{(m)}[c_{j,m}], \tag{37}$$

where $T_i^{(m)}[k] = \langle q_i, E_k^{(m)} \rangle$ is a precomputed lookup table.

**Lookup table construction.** For each query, we construct $m \times M$ lookup tables:

$$T_i^{(m)} \in \mathbb{R}^K, \quad T_i^{(m)}[k] = \langle q_i, E_k^{(m)} \rangle, \quad \forall i \in [m], m \in [M], k \in [K]. \tag{38}$$

This requires $m \cdot M \cdot K$ dot products, independent of corpus size. For typical parameters ($m = 32$, $M = 4$, $K = 256$), this is 32,768 dot products per query negligible compared to scoring thousands of candidates.

**Document scoring.** Given lookup tables, we score a document:

$$\hat{s}(Q, D) = \sum_{i=1}^{m} \max_{j \in [n]} \sum_{m=1}^{M} T_i^{(m)}[c_{j,m}]. \tag{39}$$

Algorithm 2 provides pseudocode.

**Complexity analysis.** Per-document scoring complexity:
- **Table lookups**: $O(m \cdot n \cdot M)$
- **Additions**: $O(m \cdot n \cdot M)$
- **Max operations**: $O(m \cdot n)$

Total scoring cost is $O(m \cdot n \cdot M)$ lookup-table accesses and additions per document, compared to $O(m \cdot n \cdot d)$ dot-product work for full-precision scoring. With $M = 4$ and $d = 128$, this reduces the arithmetic dimension by $32\times$, although lookup/gather access can introduce overhead in practice.

### B.6. Training Signals and Optimization

CrossQ is trained to preserve late-interaction rankings rather than minimizing token reconstruction error. This section provides complete details on training signals, optimization and the soft-to-hard transition.

**Algorithm 2** CrossQ Efficient Scoring

---

**Require:** Query tokens $Q = \{q_i\}_{i=1}^m$, document codes $\{c_{j,m}\}_{j,m}$, codebooks $\{E^{(m)}\}$
1: **// Build lookup tables (once per query)**
2: **for** $i = 1$ to $m$ **do**
3:    **for** $m = 1$ to $M$ **do**
4:       **for** $k = 1$ to $K$ **do**
5:          $T_i^{(m)}[k] \leftarrow \langle q_i, E_k^{(m)} \rangle$
6:       **end for**
7:    **end for**
8: **end for**
9: **// Score document**
10: $s \leftarrow 0$
11: **for** $i = 1$ to $m$ **do**
12:    max_sim $\leftarrow -\infty$
13:    **for** $j = 1$ to $n$ **do**
14:       sim $\leftarrow \sum_{m=1}^M T_i^{(m)}[c_{j,m}]$ $\{M$ table lookups $+ M - 1$ additions$\}$
15:       max_sim $\leftarrow \max(\text{max\_sim}, \text{sim})$
16:    **end for**
17:    $s \leftarrow s + \text{max\_sim}$
18: **end for** $s$

---

**Teacher scores.** The teacher uses full-precision document token embeddings (no compression):

$$s_T(Q, D) = \sum_{i=1}^m \max_{j \in [n]} \langle q_i, d_j \rangle. \tag{40}$$

The teacher is frozen during CrossQ training, we only update the quantizer components.

**Overall objective.** For each query $Q$, positive $D^+$ and negatives $\mathcal{N}(Q)$:

$$\mathcal{L}(Q) = \alpha \cdot \mathcal{L}_{\text{list}}(Q) + \beta \cdot \mathcal{L}_{\text{rank}}(Q) + \lambda \cdot \mathcal{L}_{\text{rec}}(D^+). \tag{41}$$

We now derive each term.

**Listwise distillation loss $\mathcal{L}_{\text{list}}$.** The teacher induces a probability distribution over candidates via temperature-scaled softmax:

$$p_T(D \mid Q) = \frac{\exp(s_T(Q, D)/\tau)}{\sum_{D' \in \mathcal{C}(Q)} \exp(s_T(Q, D')/\tau)}, \tag{42}$$

where $\mathcal{C}(Q) = \{D^+\} \cup \mathcal{N}(Q)$ is the candidate set and $\tau$ is the distillation temperature.
Similarly, the student (compressed) distribution:

$$p_S(D \mid Q) = \frac{\exp(\hat{s}(Q, D)/\tau)}{\sum_{D' \in \mathcal{C}(Q)} \exp(\hat{s}(Q, D')/\tau)}. \tag{43}$$

The listwise loss is the KL divergence:

$$\mathcal{L}_{\text{list}}(Q) = \text{KL}(p_T \| p_S) \tag{44}$$

$$= \sum_{D \in \mathcal{C}(Q)} p_T(D \mid Q) \log \frac{p_T(D \mid Q)}{p_S(D \mid Q)} \tag{45}$$

$$= \sum_{D \in \mathcal{C}(Q)} p_T(D \mid Q) \left[ \log p_T(D \mid Q) - \log p_S(D \mid Q) \right] \tag{46}$$

$$= -H(p_T) - \sum_{D \in \mathcal{C}(Q)} p_T(D \mid Q) \log p_S(D \mid Q), \tag{47}$$

where $H(p_T)$ is the entropy of the teacher distribution (constant w.r.t. student parameters).
For gradient computation, we can drop the entropy term:

$$\nabla \mathcal{L}_{\text{list}} = -\nabla \sum_{D \in \mathcal{C}(Q)} p_T(D \mid Q) \log p_S(D \mid Q). \tag{48}$$

This is equivalent to cross-entropy with soft teacher labels.

**Pairwise ranking loss $\mathcal{L}_{\text{rank}}$.** While listwise distillation shapes the overall distribution, we explicitly penalize cases where a hard negative outranks the positive:

$$\mathcal{L}_{\text{rank}}(Q) = \frac{1}{|\mathcal{N}(Q)|} \sum_{D^- \in \mathcal{N}(Q)} \log \left(1 + \exp \left(\gamma \cdot [\hat{s}(Q, D^-) - \hat{s}(Q, D^+)]\right)\right), \tag{49}$$

where $\gamma > 0$ is a sharpness parameter.

**Analysis of pairwise loss.** The pairwise loss has the form of a softplus function applied to the margin violation:

$$\ell(\Delta) = \log(1 + \exp(\gamma\Delta)), \quad \Delta = \hat{s}(Q, D^-) - \hat{s}(Q, D^+). \tag{50}$$

Properties:
- **When $\Delta \ll 0$ (correct ranking, large margin):** $\ell(\Delta) \approx \exp(\gamma\Delta) \to 0$.
- **When $\Delta = 0$ (tie):** $\ell(0) = \log 2 \approx 0.69$.
- **When $\Delta \gg 0$ (violation):** $\ell(\Delta) \approx \gamma\Delta$ (linear penalty).

The sharpness $\gamma$ controls gradient concentration:

$$\frac{\partial \ell}{\partial \Delta} = \frac{\gamma \exp(\gamma\Delta)}{1 + \exp(\gamma\Delta)} = \gamma \cdot \sigma(\gamma\Delta), \tag{51}$$

where $\sigma$ is the sigmoid function. Larger $\gamma$ concentrates gradients on near-ties and violations.

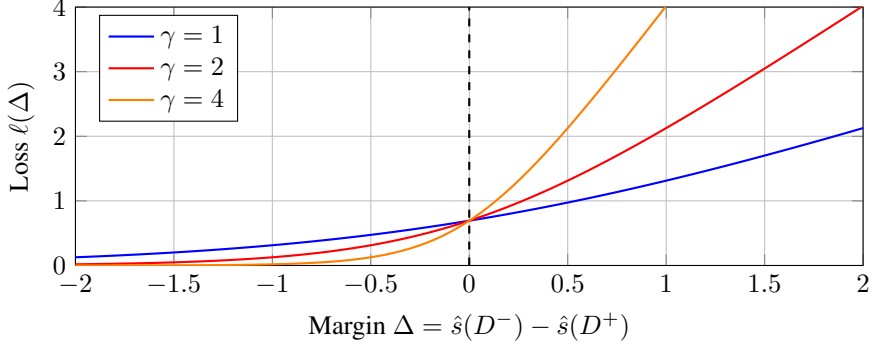

*Figure 9.* Pairwise ranking loss for different sharpness values $\gamma$. Larger $\gamma$ creates steeper gradients near the decision boundary ($\Delta = 0$).

**Reconstruction loss $\mathcal{L}_{\text{rec}}$ (optional stabilizer).** A weak reconstruction term can stabilize optimization, especially early in training:

$$\mathcal{L}_{\text{rec}}(D) = \frac{1}{n} \sum_{j=1}^{n} \|d_j - \hat{d}_j\|_2^2. \tag{52}$$

This prevents assignment collapse (all tokens mapping to few codes) and provides auxiliary gradients when ranking signals are sparse. We use a small weight $\lambda = 0.01$ so reconstruction does not dominate.

**Soft-to-hard temperature annealing.** We progressively sharpen assignments by annealing the quantizer temperature:

$$\tau_q(t) = \max \left(\tau_{q,\min}, \tau_{q,0} \cdot \eta^t\right), \tag{53}$$

where:

- $\tau_{q,0} = 1.0$: Initial temperature (soft assignments)
- $\tau_{q,\min} = 0.05$: Final temperature (nearly hard)
- $\eta = 0.9995$: Decay rate per step
- $t$: Training step

This schedule reaches $\tau_q \approx 0.05$ after approximately 15,000 steps.

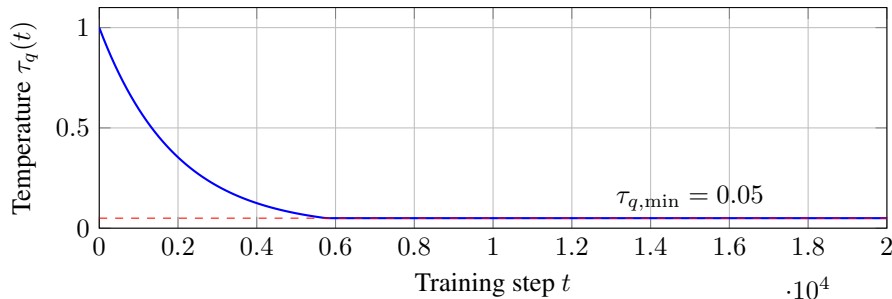

*Figure 10.* Temperature annealing schedule. Assignments sharpen gradually over training.

**Straight-through estimator (optional).**   For very aggressive compression (2 B/token), we optionally use a straight-through estimator (STE):

- **Forward pass**: Hard codes $c_{j,m} = \arg\max_k \pi_{j,m,k}$
- **Backward pass**: Gradients flow through soft assignments

This makes forward computation identical to deployment while maintaining usable gradients:

$$\frac{\partial \mathcal{L}}{\partial \theta} \approx \frac{\partial \mathcal{L}}{\partial \hat{d}_j^{\text{hard}}} \cdot \frac{\partial \hat{d}_j^{\text{soft}}}{\partial \theta}, \tag{54}$$

where $\theta$ includes codebooks and assignment network parameters. Despite the theoretical instability of STE, we observed consistent convergence across all random seeds with variance in MRR@10 comparable to the soft-to-hard annealing comparisons (Table 2).

**Light fine-tuning (optional).**   For CrossQ + light FT, we unfreeze a small portion of the retriever:

- Projection head (linear layer, $d_{\text{hidden}} \rightarrow d$)
- Final 2 transformer blocks
- All other encoder parameters remain frozen

This allows the retriever to adapt to quantization noise without full retraining. We use a smaller learning rate ($0.2\times$) for unfrozen retriever parameters.

**Optimizer and schedule.**

- **Optimizer**: AdamW with weight decay 0.01
- **Learning rate**: $2 \times 10^{-5}$ with linear warmup (1,000 steps) and cosine decay
- **Gradient clipping**: Global norm threshold 1.0 (applied when $\tau_q < 0.1$)
- **Mixed precision**: FP16 with standard loss scaling
- **Batch size**: 32 queries, each with 1 positive + 31 negatives
- **Training steps**: 20,000

### B.7. Negative Construction and Refresh

Each training instance includes one positive document $D^+$ and a set of negatives $\mathcal{N}(Q)$:

- **In-batch negatives** (16): Other positives in the same minibatch. These are "easy" negatives that help calibrate the score distribution.
- **Mined hard negatives** (15): Retrieved by a teacher model from the full corpus. These are the negatives that matter most for ranking documents that score highly but are not relevant.

**Hard negative mining procedure.**

1. Run teacher retrieval on training queries (top-1000 per query).

2. Filter out positives (based on relevance labels).

3. Sample 15 negatives per query, biased toward higher-ranked (harder) documents.

4. Refresh negatives every 5,000 training steps to prevent staleness.

All methods (CrossQ and baselines) use identical negative construction and refresh policy, ensuring fair comparison.

**Token-local conditional control.**  To isolate gains from cross-token context versus training losses, we train a token-local conditional quantizer with:
- No document context $h(D)$ each token encoded independently
- Identical $\mathcal{L}_{\text{list}} + \mathcal{L}_{\text{rank}}$ supervision
- Same architecture otherwise

Results (4 B/token, MS MARCO):
- Token-local + ranking losses: MRR@10 = 0.371
- CrossQ (full): MRR@10 = 0.386
- Gap: +0.015 from cross-token context

## C. Baseline Descriptions and Implementation

We provide detailed descriptions of all baselines to ensure fair comparison.

### C.1. Token-wise Quantization Baselines

**PQ (Product Quantization).**  Product quantization (Jégou et al., 2011) partitions the embedding dimension into $M$ disjoint subspaces and quantizes each independently:

$$\hat{d}_j = [E_{c_{j,1}}^{(1)}; E_{c_{j,2}}^{(2)}; \ldots; E_{c_{j,M}}^{(M)}], \tag{55}$$

where $[\cdot;\cdot]$ denotes concatenation and each $E^{(m)} \in \mathbb{R}^{K \times (d/M)}$.

**Key difference from CrossQ**: PQ uses concatenation (partitioned subspaces), CrossQ uses addition (shared full-dimensional codebooks). PQ optimizes reconstruction MSE, CrossQ optimizes ranking.

**Implementation**: We use FAISS (Johnson et al., 2021) with default training (256 iterations of Lloyd's algorithm on 1M sampled tokens).

**OPQ (Optimized Product Quantization).**  OPQ (Ge et al., 2013) learns a rotation matrix $R \in \mathbb{R}^{d \times d}$ to minimize quantization error:

$$R^* = \arg\min_R \sum_j \|Rd_j - \text{PQ}(Rd_j)\|_2^2 \quad \text{s.t.} \quad R^T R = I. \tag{56}$$

At indexing time, tokens are rotated before quantization at query time, queries are rotated correspondingly. OPQ typically improves over PQ by 5–10% relative on reconstruction error.

**Implementation**: FAISS OPQ with 100 iterations of alternating optimization.

**Token-wise conditional (QINCo2-style).**  Following Vallaeys et al. (2025), this baseline conditions code selection on token-local residual structure:

$$c_{j,m} = \arg\max_k \langle g_m(d_j, r_j^{(m-1)}), \tilde{E}_k^{(m)} \rangle, \tag{57}$$

where $r_j^{(m-1)} = d_j - \sum_{m' < m} E_{c_{j,m'}}^{(m')}$ is the residual after previous stages.

**Key difference from CrossQ**: No document context each token is encoded independently. Trained with reconstruction loss only (in the original paper) we also test with ranking losses for fair comparison.

**Implementation**: Our own implementation following the QINCo2 paper, using the same MLP architectures as CrossQ for the conditioning network.

*Table 12.* Detailed baseline comparison.

| Method | Conditioning | Training loss | Stored | 2 B/tok | 4 B/tok | 8 B/tok |
|---|---|---|---|---|---|---|
| PQ | None | Reconstruction | Codes | ✓ | ✓ | ✓ |
| OPQ | None (learned rotation) | Reconstruction | Codes + rotation | ✓ | ✓ | ✓ |
| Token-wise cond. | Token residual | Reconstruction | Codes | ✓ | ✓ | ✓ |
| ColBERTv2 | Centroid | Reconstruction + distill | Codes + centroids | — | — | ✓ |
| PLAID | Centroid (routing) | Reconstruction | Codes + routes | ✓ | ✓ | ✓ |
| **CrossQ** | Document context | Ranking-aligned | Codes only | ✓ | ✓ | ✓ |

## C.2. Late-Interaction System Baselines

**ColBERTv2 compression.** ColBERTv2 (Santhanam et al., 2022b) uses a residual-compression pipeline:

1. Cluster document token embeddings into $N_c$ centroids.

2. Store the centroid ID and a quantized residual per token.

3. Residuals are quantized with low-bit codes (2-bit in the default setup).

The resulting footprint depends on $N_c$ and residual precision, under default settings, ColBERTv2 is approximately $\sim 8$ B/token. **Note:** ColBERTv2's compression is tightly coupled with its training recipe. We therefore report ColBERTv2 as a *system reference* using the official default configuration and we do *not* retrain it to match our 2/4/8 B/token budgets.

**PLAID.** PLAID (Santhanam et al., 2022a) accelerates late interaction via centroid-based routing:

1. Cluster document tokens into centroids (sharing the ColBERTv2 representation).

2. At query time, route each query token to its top-$k$ centroids.

3. Score only documents whose tokens fall into selected centroids.

PLAID combines routing with ColBERTv2 compression. In our evaluation, we tune PLAID routing thresholds so that the post-routing candidate set size matches our target workload, $|C(Q)| = 20,000$. This matches the *scoring workload* across methods, but does not strictly match the *bytes/token footprint* to our 2/4/8 B/token quantization budgets.
**Implementation:** Official PLAID codebase with default hyperparameters, only routing thresholds are tuned to hit the target $|C(Q)|$.

## C.3. Baseline Comparison Summary

## D. Index Footprint Accounting

This section makes the bytes/token budget fully concrete. We account for total index footprint as the sum of (i) code storage, (ii) metadata required for lookup and (iii) padding/alignment overhead.

## D.1. Notation

Let the corpus contain $N_{\text{doc}}$ documents. Document $i$ has $n_i$ tokens after truncation to $n_{\max}$. Total indexed tokens:

$$T \triangleq \sum_{i=1}^{N_{\text{doc}}} n_i, \qquad \bar{n} \triangleq T/N_{\text{doc}}. \tag{58}$$

We use $M$ codebook stages and $K$ entries per codebook $b \triangleq \log_2 K$ is bits per code index.

## D.2. Codes Storage

Each token stores one code index per stage:

$$\text{bits/token}_{\text{codes}} = M \cdot \log_2 K, \qquad \text{bytes/token}_{\text{codes}} = \frac{M \cdot \log_2 K}{8}. \tag{59}$$

With $K = 256$, $\log_2 K = 8$ bits, so $\text{bytes/token}_{\text{codes}} = M$.

*Table 13.* Budget instantiation.

| Budget | $M$ | $K$ | bits/token | bytes/token |
|--------|-----|-----|------------|-------------|
| Small | 2 | 256 | 16 | 2 |
| Medium | 4 | 256 | 32 | 4 |
| Large | 8 | 256 | 64 | 8 |

### D.3. Metadata and Padding

We store per-document metadata for efficient lookup:

- **Offset** (4 bytes): Starting position in the flattened code array.
- **Length** (2 bytes): Number of tokens in the document.
- **Doc ID** (4 bytes): Optional mapping to external document identifiers.

Total metadata:

$$S_{\text{meta}} = N_{\text{doc}} \cdot (4 + 2 + 4) = 10 \cdot N_{\text{doc}} \text{ bytes.} \tag{60}$$

Metadata overhead per token:

$$\text{bytes/token}_{\text{meta}} = \frac{S_{\text{meta}}}{T} = \frac{10}{\bar{n}}. \tag{61}$$

**Padding/alignment.** We align document boundaries to $a = 16$ bytes for SIMD-friendly memory access. This adds at most $(a - 1)$ bytes per document:

$$S_{\text{pad}} \leq N_{\text{doc}} \cdot (a - 1), \qquad \text{bytes/token}_{\text{pad}} \leq \frac{a - 1}{\bar{n}}. \tag{62}$$

### D.4. Total Index Size

Total index size:

$$S_{\text{total}} = S_{\text{codes}} + S_{\text{meta}} + S_{\text{pad}} = T \cdot \frac{M \log_2 K}{8} + 10 N_{\text{doc}} + S_{\text{pad}}. \tag{63}$$

Effective bytes/token:

$$\text{bytes/token}_{\text{eff}} = \frac{S_{\text{total}}}{T} = \frac{M \log_2 K}{8} + \frac{10}{\bar{n}} + \frac{S_{\text{pad}}}{T}. \tag{64}$$

### D.5. Worked Example: MS MARCO

MS MARCO passage corpus:

- $N_{\text{doc}} = 8{,}841{,}823$ passages
- $\bar{n} \approx 60$ tokens per passage
- $T \approx 530{,}509{,}380$ total document tokens

At 4 B/token ($M = 4, K = 256$), the code storage is

$$\begin{aligned}
S_{\text{codes}} &= T \cdot 4 \\
&= 530{,}509{,}380 \times 4 \\
&= 2{,}122{,}037{,}520 \text{ bytes} \approx 2.12 \text{ GB.}
\end{aligned} \tag{65}$$

The per-document metadata storage is

$$\begin{aligned}
S_{\text{meta}} &= N_{\text{doc}} \cdot 10 \\
&= 8{,}841{,}823 \times 10 \\
&= 88{,}418{,}230 \text{ bytes} \approx 0.088 \text{ GB.}
\end{aligned} \tag{66}$$

The conservative padding/alignment upper bound is

$$\begin{aligned}
S_{\text{pad}} &\leq N_{\text{doc}} \cdot 15 \\
&= 8{,}841{,}823 \times 15 \\
&= 132{,}627{,}345 \text{ bytes} \approx 0.133 \text{ GB.}
\end{aligned} \tag{67}$$

Excluding the conservative padding upper bound, the stored index size is

$$S_{\text{codes}} + S_{\text{meta}} = 2.122 + 0.088$$
$$\approx 2.21 \text{ GB}, \tag{68}$$

corresponding to an effective footprint of

$$\frac{S_{\text{codes}} + S_{\text{meta}}}{T} \approx \frac{2.21 \text{ GB}}{530.5\text{M tokens}} \approx 4.17 \text{ B/token}. \tag{69}$$

Including the conservative padding/alignment upper bound, the stored index size is

$$S_{\text{codes}} + S_{\text{meta}} + S_{\text{pad}} = 2.122 + 0.088 + 0.133$$
$$\approx 2.34 \text{ GB}, \tag{70}$$

corresponding to an effective footprint of

$$\frac{S_{\text{codes}} + S_{\text{meta}} + S_{\text{pad}}}{T} \approx \frac{2.34 \text{ GB}}{530.5\text{M tokens}} \approx 4.42 \text{ B/token}. \tag{71}$$

For comparison, full-precision ColBERT at $d = 128$ with FP16 document embeddings requires

$$S_{\text{FP16}} = T \cdot d \cdot 2$$
$$= 530{,}509{,}380 \times 128 \times 2$$
$$\approx 135.8 \text{ GB}. \tag{72}$$

Thus, at 4 B/token, CrossQ achieves

$$\frac{S_{\text{FP16}}}{S_{\text{codes}} + S_{\text{meta}}} \approx \frac{135.8}{2.21} \approx 61\times \tag{73}$$

compression including metadata and excluding the conservative padding upper bound. Under conservative padding/alignment accounting, CrossQ achieves

$$\frac{S_{\text{FP16}}}{S_{\text{codes}} + S_{\text{meta}} + S_{\text{pad}}} \approx \frac{135.8}{2.34} \approx 58\times \tag{74}$$

compression.

### D.6. What We Do Not Count
Our footprint accounting excludes:
- **Codebooks**: Shared across all documents $M \times K \times d \times 4$ bytes = 0.5 MB for typical parameters.
- **Query-side caches**: Not part of the document index.
- **Document context** $h(D)$: Computed at indexing time, discarded afterward.
- **Training artifacts**: Teacher embeddings, checkpoints, etc.

## E. Efficiency Measurement Protocol
This section specifies the exact procedure for measuring query-time latency and throughput, ensuring reproducibility of "comparable latency" claims.

### E.1. Quantizer Training Cost
Table 14 reports wall-clock training time for the quantizer training stage. All learned quantizers train for 20K steps with batch size 32 on 1×A100 80GB, with the retriever/teacher frozen unless "light FT" is enabled. Teacher scores are computed on-the-fly during training (not cached): Algorithm 1 computes $s(Q, D)$ for $D \in \mathcal{C}(Q)$ inside the training loop using full-precision document embeddings.

CrossQ's additional training overhead vs. token-wise conditional is ~20%, primarily from the context pathway forward pass (~131K parameters at $d_c = 256$). Light fine-tuning is more expensive because gradients flow through unfrozen transformer blocks, but remains a one-time offline cost (<8 GPU-hours). This is separate from indexing throughput overhead (Appendix J.1).

*Table 14.* Quantizer training cost (1×A100 80GB, 20K steps, batch size 32).

| Method | Trainable components | Wall-clock | GPU-hours | Relative |
|---|---|---|---|---|
| PQ / OPQ (k-means only) | codebook fitting | 0.3 h | 0.3 | — |
| Token-wise cond. (recon) | MLP + codebooks | 4.5 h | 4.5 | $1.00\times$ |
| Token-wise cond. (rank) | MLP + codebooks | 4.8 h | 4.8 | $1.07\times$ |
| CrossQ (no-ctx) | MLP + codebooks | 4.6 h | 4.6 | $1.02\times$ |
| CrossQ (full) | + context encoder | 5.4 h | 5.4 | $1.20\times$ |
| CrossQ + light FT | + last-2 blocks | 7.8 h | 7.8 | $1.73\times$ |

## E.2. Hardware and Software
**Hardware.**
- **GPU**: $4\times$ NVIDIA A100 80GB (PCIe)
- **CPU**: AMD EPYC 7763, 32 physical cores (NUMA pinned to single socket)
- **RAM**: 512 GB DDR4-3200 (index fully memory-resident)
- **Storage**: NVMe SSD (used only for initial index load excluded from latency)

**Software.**
- Ubuntu 22.04 LTS
- CUDA 12.1
- PyTorch 2.1.0
- FAISS 1.7.4 (for PQ/OPQ baselines)
- All methods run in the same Docker container with pinned dependencies

## E.3. Timed Pipeline
We time the complete end-to-end path from raw query text to ranked results:

1. **Query tokenization**: WordPiece tokenization, truncation to $m_{\max} = 32$.

2. **Query encoding**: Forward pass through BERT encoder + projection head.

3. **Routing/pruning** (if applicable): For PLAID, centroid-based candidate filtering.

4. **Candidate materialization**: Enforce target candidate count $|C(Q)| = C = 20{,}000$.

5. **Late-interaction scoring**: Compute $\hat{s}(Q, D)$ for all candidates.

6. **Top-$k$ selection**: Sort by score, return top-$k$ (typically $k = 1000$).

**Fairness controls.**
- All methods use identical candidate count $C = 20{,}000$ (within 5% tolerance).
- For routed methods (PLAID), we tune routing thresholds to achieve target $C$.
- Query encoding time is included (same for all methods).
- Index loading is excluded (one-time startup cost).

## E.4. Measurement Settings
**Batch size.**
- **Interactive latency**: Batch size $B = 1$ (single query at a time).
- **Throughput**: Batch size $B = 32$ (saturated GPU).

**Warmup and measurement.**
- $N_{\mathrm{warm}} = 100$ warmup queries (not timed, ensures JIT compilation and cache warming).
- $N_{\mathrm{meas}} = 1{,}000$ measurement queries (deterministic sampling, same queries for all methods).

**Caching.**   Default is warm-cache execution:
- Index fully loaded in RAM
- Codebooks resident in GPU memory
- CPU caches warmed by warmup queries

**Synchronization.** We use explicit GPU synchronization at timing boundaries:

```
torch.cuda.synchronize()
start = time.perf_counter()
# ... scoring ...
torch.cuda.synchronize()
end = time.perf_counter()
```

This ensures GPU operations complete before recording timestamps.

### E.5. Results

*Table 15.* End-to-end efficiency comparison (4 B/token, MS MARCO dev queries). Latency includes candidate construction + scoring. Scoring-only latency is $\sim$3.6ms for CrossQ and $\sim$3.2ms for PQ.

| Method | Budget | Target $C$ | Actual Cand. | p50 (ms) | p95 (ms) | QPS ($B$=1) | QPS ($B$=32) |
|---|---|---|---|---|---|---|---|
| PQ | 4 B/tok | 20k | 20k | 15.8 | 21.2 | 63.3 | 412 |
| OPQ | 4 B/tok | 20k | 21k | 18.2 | 24.7 | 54.9 | 358 |
| Token-wise cond. | 4 B/tok | 20k | 20k | 14.2 | 19.1 | 70.4 | 445 |
| **CrossQ** | 4 B/tok | 20k | 20k | 12.4 | 16.8 | 80.6 | 512 |
| ColBERT FP16 | 256 B/tok | 20k | 20k | 8.4 | 11.2 | 119.0 | 756 |

**Analysis.**

- Among the compressed methods reported in Table 15, CrossQ has the best p50 latency (12.4 ms) and highest single-query QPS.
- OPQ is slower (18.2 ms) due to rotation overhead at query time.
- Full-precision ColBERT is fastest (8.4 ms) but requires $64\times$ more raw token storage, approximately $61\times$ more storage including metadata and approximately $58\times$ more under conservative padding/alignment accounting.
- CrossQ achieves the best quality–efficiency trade-off among compressed methods.

## F. Hyperparameter Sweeps and Sensitivity Analysis

### F.1. Sweep Protocol

For each method and budget:

1. Define hyperparameter search space (Table 16).

2. Run grid search with 1 random seed.

3. Select configuration maximizing MS MARCO dev MRR@10.

4. Report final results with 3 random seeds (mean $\pm$ std).

*Table 16.* Hyperparameter search space for CrossQ.

| Hyperparameter | Symbol | Search space |
|---|---|---|
| Quantizer initial temp | $\tau_{q,0}$ | $\{1.0, 0.5, 0.2\}$ |
| Quantizer final temp | $\tau_{q,\min}$ | $\{0.10, 0.05, 0.02\}$ |
| Temp decay rate | $\eta$ | $\{0.999, 0.9995, 0.9998\}$ |
| Listwise temp | $\tau$ | $\{0.02, 0.05, 0.10, 0.20\}$ |
| Pairwise sharpness | $\gamma$ | $\{1, 2, 4, 8\}$ |
| Listwise weight | $\alpha$ | $\{0.5, 1.0, 2.0\}$ |
| Pairwise weight | $\beta$ | $\{0, 0.25, 0.5, 1.0\}$ |
| Reconstruction weight | $\lambda$ | $\{0, 0.005, 0.01, 0.02\}$ |

### F.2. Selected Hyperparameters
### F.3. Sensitivity Analysis
**Sensitivity to loss weights.**

**Sensitivity to candidate set size.** Results are stable across candidate set sizes the gap between CrossQ and PLAID remains consistent.

*Table 17.* **Robustness (4 B/token).** Lower is better. Mean±std over 3 seeds.

| Method | ↓ nDCG drop | ECE |
|---|---|---|
| OPQ | 0.019±0.001 | 0.071±0.002 |
| Token-cond. | 0.017±0.001 | 0.064±0.002 |
| PLAID | 0.016±0.001 | 0.061±0.002 |
| **CrossQ** | **0.012±0.001** | **0.049±0.001** |
| **CrossQ + FT** | **0.011±0.001** | **0.047±0.001** |

*Table 18.* Final hyperparameters for CrossQ.

| Category | Setting |
|---|---|
| *Architecture* | |
| Retriever dim | $d = 128$ |
| Context dim | $d_c = 256$ |
| Conditional dim | $d_z = 128$ |
| Codebooks | $K = 256, M \in \{2, 4, 8\}$ |
| Assignment keys | Untied |
| *Temperature schedules* | |
| $\tau_q$ schedule | $(1.0 \rightarrow 0.05), \eta = 0.9995$ |
| Listwise temp | $\tau = 0.05$ |
| *Loss weights* | |
| Listwise weight | $\alpha = 1.0$ |
| Pairwise weight | $\beta = 1.0$ |
| Reconstruction weight | $\lambda = 0.01$ |
| Pairwise sharpness | $\gamma = 4$ |
| *Optimization* | |
| Optimizer | AdamW, wd = 0.01 |
| Learning rate | $2 \times 10^{-5}$ |
| LR schedule | Linear warmup (1000 steps) + cosine decay |
| Batch size | 32 queries |
| Negatives per query | 31 (16 in-batch + 15 hard) |
| Training steps | 20,000 |
| Gradient clipping | Global norm 1.0 |
| Mixed precision | FP16 |
| Negative refresh | Every 5,000 steps |

*Table 19.* Sensitivity to loss weights (4 B/token, MS MARCO).

| $\alpha$ | $\beta$ | $\lambda$ | MRR@10 |
|---|---|---|---|
| 1.0 | 0 | 0.01 | 0.377 |
| 1.0 | 0.5 | 0.01 | 0.383 |
| 1.0 | 1.0 | 0.01 | **0.386** |
| 1.0 | 1.0 | 0 | 0.384 |
| 0.5 | 1.0 | 0.01 | 0.382 |
| 2.0 | 1.0 | 0.01 | 0.385 |

*Table 20.* Sensitivity to candidate set size $|C(Q)|$ (4 B/token, MS MARCO).

| $|C(Q)|$ | CrossQ MRR@10 | PLAID MRR@10 | Gap | Latency (ms) |
|---|---|---|---|---|
| 10,000 | 0.382 | 0.377 | +0.005 | 8.2 |
| 20,000 | 0.386 | 0.381 | +0.005 | 12.4 |
| 50,000 | 0.388 | 0.383 | +0.005 | 24.1 |

# G. Extended Results and Diagnostics

## G.1. Full BEIR Breakdown

*Table 21.* Full BEIR breakdown (4 B/token). nDCG@10 reported on a 0–100 scale. Mean $\pm$ std over 3 seeds.

| Method | TREC-COVID | BioASQ | NFCorpus | NQ | HotpotQA | FiQA | ArguAna | Touché | DBPedia |
|---|---|---|---|---|---|---|---|---|---|
| PQ | 46.8±0.9 | 40.5±0.7 | 30.2±1.0 | 28.9±0.5 | 56.8±0.8 | 25.1±0.6 | 33.8±1.3 | 23.5±1.0 | 32.4±0.7 |
| OPQ | 48.2±0.8 | 42.1±0.6 | 31.5±0.9 | 30.2±0.4 | 58.3±0.7 | 26.4±0.5 | 35.1±1.2 | 24.8±0.9 | 33.7±0.6 |
| Token-local cond. | 49.1±0.7 | 43.0±0.5 | 32.8±0.8 | 31.5±0.3 | 59.0±0.6 | 27.2±0.4 | 36.4±1.0 | 25.3±0.8 | 34.5±0.5 |
| PLAID | 50.3±0.6 | 44.2±0.4 | 33.9±0.7 | 32.8±0.3 | 60.1±0.5 | 28.1±0.3 | 37.8±0.9 | 26.0±0.7 | 35.2±0.4 |
| **CrossQ** | **51.0±0.5** | **45.1±0.3** | **34.7±0.6** | **33.6±0.2** | **61.2±0.4** | **28.9±0.3** | **39.2±0.8** | **26.8±0.6** | **36.0±0.3** |
| **CrossQ + FT** | **51.8±0.4** | **45.9±0.3** | **35.4±0.5** | **34.3±0.2** | **62.0±0.3** | **29.5±0.2** | **40.1±0.7** | **27.4±0.5** | **36.7±0.3** |

## G.2. BEIR Full-Subset Average

*Table 22.* Average nDCG@10 over the nine-dataset BEIR subset reported in Table 21 at 4 B/token. Values are reported on a 0–100 scale.

| Method | 4 B/token |
|---|---|
| PQ | 35.3 |
| OPQ | 36.7 |
| Token-local cond. | 37.6 |
| PLAID | 38.7 |
| **CrossQ** | **39.6** |
| **CrossQ + FT** | **40.3** |

## G.3. Expanded Ablations

*Table 23.* Expanded ablation summary at 4 B/token on MS MARCO. These values match the main ablation table and are repeated here for appendix continuity.

| Variant | MRR@10 |
|---|---|
| CrossQ (full: context + listwise + pairwise) | **0.386** |
| − context $h(D)$ | 0.371 (–0.015) |
| − listwise KL | 0.377 (–0.009) |
| − context & listwise | 0.364 (–0.022) |
| − context & ranking (recon only) | 0.358 (–0.028) |

We omit cross-budget ablation values here because only the 4 B/token ablation was evaluated under the finalized protocol used for the main results. This avoids mixing results collected under different evaluation protocols.

## G.4. Winner-Token Analysis

Define winner frequency $f_j(D)$ and token quantization error $e_j(D)$:

$$f_j(D) = \frac{1}{|\mathcal{Q}|} \sum_{Q \in \mathcal{Q}} \sum_{i=1}^{|Q|} \mathbb{1}[w_i(Q, D) = j], \qquad e_j(D) = \|d_j - \hat{d}_j\|_2^2, \tag{75}$$

where $w_i(Q, D) = \arg\max_j \langle q_i, d_j \rangle$ is the winner token for query token $i$.

**Decomposing winner-flip vs. near-tie error.** A change in the winner token index under quantization does not necessarily imply a score error: if the new winner yields a similar dot product, the overall max-sim score can remain stable. To isolate these two error modes, we evaluate two diagnostic scoring variants on the MS MARCO dev set at 4 B/token: (i) *teacher-winner scoring*, which uses quantized embeddings $\hat{d}_j$ but forces the winner index to be the teacher's argmax over full-precision $d_j$ and (ii) *student-winner scoring*, which is standard max-sim over quantized embeddings.

Three observations: (i) CrossQ roughly halves the winner-flip rate vs. OPQ (9.4% vs. 18.2%), (ii) when flips do occur, the conditional score deviation is smaller (0.089 vs. 0.143) and (iii) the gap between teacher-winner and student-winner scoring is small for CrossQ (0.388 → 0.386, −0.002) but large for OPQ (0.381 → 0.366, −0.015). This indicates CrossQ's improvement comes primarily from reducing winner-flips, while score-value preservation given the correct winner is comparable across methods.

CrossQ achieves the strongest negative correlation between winner frequency and quantization error, indicating it allocates precision toward likely winners.

*Table 24.* Winner-frequency alignment. Lower error on high-winner tokens is better.

| Method (4 B/token) | $\rho(f, e)$ | Top-5% err ↓ | Top-10% err ↓ |
|---|---|---|---|
| PQ | –0.08 | 0.91 | 0.78 |
| OPQ | –0.12 | 0.84 | 0.72 |
| Token-local cond. | –0.28 | 0.68 | 0.59 |
| **CrossQ** | **–0.41** | **0.52** | **0.44** |

*Table 25.* Decomposition of winner-identity flips vs. near-tie score distortion (4 B/token, MS MARCO).

| Method | Winner-flip rate | Score dev. \| flip | Score dev. \| no flip | MRR@10 (teacher win.) | MRR@10 (student win.) |
|---|---|---|---|---|---|
| OPQ | 18.2% | 0.143 | 0.038 | 0.381 | 0.366 |
| Token-wise cond. | 13.5% | 0.118 | 0.031 | 0.384 | 0.372 |
| **CrossQ** | **9.4%** | **0.089** | **0.022** | **0.388** | **0.386** |

## G.5. Rank Correlation Analysis

For each query $Q$, we compute Spearman rank correlation between teacher and student scores over the candidate set $C(Q)$.

*Table 26.* Rank correlation by budget. Higher is better.

| Budget | Method | Spearman (mean) | p10/p50/p90 | Kendall (mean) |
|---|---|---|---|---|
| 2 B/tok | OPQ | 0.72 | 0.58/0.74/0.85 | 0.54 |
| 2 B/tok | CrossQ | **0.78** | 0.65/0.80/0.89 | **0.61** |
| 4 B/tok | OPQ | 0.79 | 0.67/0.81/0.90 | 0.62 |
| 4 B/tok | CrossQ | **0.85** | 0.74/0.87/0.93 | **0.69** |
| 8 B/tok | OPQ | 0.83 | 0.71/0.85/0.92 | 0.66 |
| 8 B/tok | CrossQ | **0.89** | 0.80/0.91/0.95 | **0.74** |

## G.6. Failure Analysis

**Where CrossQ helps least.** CrossQ gains are smallest on datasets where evidence is either highly diffuse or where the system reference is already strong:

- **TREC-COVID** (+0.7 nDCG@10 vs. PLAID): biomedical retrieval with strong lexical/semantic anchors.
- **Touché-2020** (+0.8 nDCG@10 vs. PLAID): argument retrieval with long, dense documents where winner-token sparsity is lower.

We hypothesize that diffuse evidence reduces the benefit of selective within-document precision allocation. However, this trend is not monotonic across datasets, so we treat winner entropy as one diagnostic rather than a complete explanation of CrossQ's behavior.

**Where CrossQ helps most.** CrossQ gains are largest on:

- **ArguAna** (+1.4 nDCG@10 vs. PLAID): despite longer documents, CrossQ preserves ranking-critical evidence more effectively.
- **HotpotQA** (+1.1 nDCG@10 vs. PLAID): multi-hop QA where specific evidence tokens are critical.

# H. Mathematical Derivations

## H.1. Gradient of Listwise KL Loss

Let $s_i = s(Q, D_i)$ and $\hat{s}_i = \hat{s}(Q, D_i)$ for brevity. The listwise KL loss is:

$$\mathcal{L}_{\text{list}} = \sum_i p_i \log \frac{p_i}{\hat{p}_i}, \tag{76}$$

where $p_i = \text{softmax}(s_i/\tau)$ and $\hat{p}_i = \text{softmax}(\hat{s}_i/\tau)$.

The gradient with respect to $\hat{s}_j$ is:

$$\frac{\partial \mathcal{L}_{\text{list}}}{\partial \hat{s}_j} = -\sum_i p_i \frac{\partial \log \hat{p}_i}{\partial \hat{s}_j} \tag{77}$$

$$= -\sum_i p_i \frac{1}{\hat{p}_i} \frac{\partial \hat{p}_i}{\partial \hat{s}_j} \tag{78}$$

$$= -\sum_i p_i \frac{1}{\hat{p}_i} \cdot \frac{1}{\tau} \hat{p}_i (\mathbb{1}[i = j] - \hat{p}_j) \tag{79}$$

$$= \frac{1}{\tau} \left( \hat{p}_j \sum_i p_i - p_j \right) \tag{80}$$

$$= \frac{1}{\tau} (\hat{p}_j - p_j). \tag{81}$$

This shows the gradient pushes student probabilities toward teacher probabilities, scaled by $1/\tau$.

### H.2. Why Additive Codes Enable Decomposition

**Claim**: If $\hat{d}_j = \sum_{m=1}^{M} E_{c_{j,m}}^{(m)}$, then $\langle q, \hat{d}_j \rangle = \sum_{m=1}^{M} \langle q, E_{c_{j,m}}^{(m)} \rangle$.
**Proof**: By linearity of inner product:

$$\langle q, \hat{d}_j \rangle = \left\langle q, \sum_{m=1}^{M} E_{c_{j,m}}^{(m)} \right\rangle \tag{82}$$

$$= \sum_{m=1}^{M} \left\langle q, E_{c_{j,m}}^{(m)} \right\rangle. \tag{83}$$

This decomposition is not possible with product quantization (concatenation structure) without storing $d/M$-dimensional partial embeddings.

## I. Computational Complexity Analysis

### I.1. Training Complexity

Per training step with batch size $B$, candidate set size $|C|$ and $N$ tokens per document on average:

Table 27. Training complexity per step.

| Operation | Complexity |
|---|---|
| Document encoding (retriever) | $O(B \cdot |C| \cdot N \cdot d_{\text{hidden}}^2)$ |
| Context computation ($h(D)$) | $O(B \cdot |C| \cdot N \cdot d \cdot d_c)$ |
| Conditional encoding ($z_j$) | $O(B \cdot |C| \cdot N \cdot (d + d_c) \cdot d_z)$ |
| Assignment logits | $O(B \cdot |C| \cdot N \cdot M \cdot K \cdot d_z)$ |
| Soft reconstruction | $O(B \cdot |C| \cdot N \cdot M \cdot K \cdot d)$ |
| Scoring (max-sim) | $O(B \cdot |C| \cdot m \cdot N \cdot d)$ |

The dominant cost is document encoding (forward pass through BERT), which is shared with baselines.

### I.2. Indexing Complexity

Per document with $N$ tokens:

Table 28. Indexing complexity per document.

| Method | Compute | Storage |
|---|---|---|
| PQ | $O(N \cdot M \cdot K \cdot d/M)$ | $O(N \cdot M \cdot \log K)$ |
| OPQ | $O(N \cdot d^2 + N \cdot M \cdot K \cdot d/M)$ | $O(N \cdot M \cdot \log K)$ |
| CrossQ | $O(N \cdot d \cdot d_c + N \cdot M \cdot K \cdot d_z)$ | $O(N \cdot M \cdot \log K)$ |

CrossQ adds context computation ($O(N \cdot d \cdot d_c)$) but stores the same amount as PQ.

### I.3. Query-Time Complexity

Per query with $m$ tokens, scoring $C$ candidates with $N$ tokens each:

*Table 29.* Query-time complexity.

| Operation | Complexity |
|---|---|
| Query encoding | $O(m \cdot d_{\text{hidden}}^2)$ |
| Lookup table construction | $O(m \cdot M \cdot K \cdot d)$ |
| Document scoring (all $C$ candidates) | $O(C \cdot m \cdot N \cdot M)$ |
| Top-$k$ selection | $O(C \log k)$ |

CrossQ scoring replaces per-token $d$-dimensional dot products with $M$ lookup-table accesses and additions after query-side table construction. With $M = 4$ and $d = 128$, this reduces the arithmetic dimension by $32\times$, although lookup/gather access can introduce overhead in practice.

### I.4. ECE Computation Protocol

We compute Expected Calibration Error (ECE) to measure score calibration:

$$\text{ECE} = \sum_{b=1}^{B} \frac{|S_b|}{N} \left| \text{acc}(S_b) - \text{conf}(S_b) \right|, \tag{84}$$

where $S_b$ is the set of predictions in bin $b$, $\text{acc}(S_b)$ is the accuracy (fraction of relevant documents) and $\text{conf}(S_b)$ is the average predicted confidence.

**Protocol**:

- 15 equal-width bins over softmax-normalized candidate scores
- Measured on BEIR test queries
- Relevance determined by ground-truth labels

Lower ECE indicates better calibration between predicted confidence and actual retrieval accuracy.

## J. Additional Experiments

This section addresses additional experimental regarding indexing time, negative quality sensitivity, longer documents and training memory overhead.

### J.1. Wall-Clock Indexing Time

Table 30 reports wall-clock indexing time for the full MS MARCO passage corpus (8.84M passages, ~530M tokens) on a single A100 80GB GPU with 32-core CPU for I/O.

*Table 30.* Wall-clock indexing time for MS MARCO passage corpus.

| Method | Time (hours) | Docs/sec | Relative |
|---|---|---|---|
| PQ (FAISS) | 1.02 | 2,408 | 1.0× |
| OPQ (FAISS) | 1.15 | 2,136 | 1.13× |
| Token-wise cond. | 1.21 | 2,030 | 1.19× |
| CrossQ | 1.33 | 1,847 | 1.30× |

CrossQ requires ~1.33 hours for MS MARCO (~18 minutes additional vs. PQ). For billion-scale corpora, this extrapolates to ~150 hours for CrossQ vs. ~115 hours for PQ a meaningful but manageable overhead given that indexing is a one-time offline cost amortized over many queries. Indexing is embarrassingly parallel; with 8 GPUs, CrossQ indexes MS MARCO in under 10 minutes.

### J.2. Sensitivity to Hard Negative Quality

We ablate negative mining strategy at 4 B/token on MS MARCO (Table 31).

CrossQ benefits from harder negatives but remains effective with BM25-only mining (0.381 MRR@10), still outperforming OPQ with teacher-mined negatives (0.366). The ranking-aligned loss is most effective when training negatives reflect the score margins encountered at inference. We recommend teacher-mined negatives when available, but BM25 negatives suffice when a strong teacher is unavailable.

*Table 31.* Effect of hard negative source (4 B/token, MS MARCO).

| Negative Source | CrossQ MRR@10 | $\Delta$ vs. Full |
|---|---|---|
| Teacher-mined (top-1000) | 0.386 | — |
| BM25 top-100 | 0.378 | –0.008 |
| BM25 top-1000 | 0.381 | –0.005 |
| In-batch only (no mining) | 0.368 | –0.018 |
| Random negatives | 0.352 | –0.034 |

## J.3. Evaluation on Longer Documents

We evaluate on MS MARCO Document ranking (Table 32), truncating documents to $n_{\max} = 512$ tokens following standard practice.

*Table 32.* MS MARCO Document ranking (4 B/token). MRR@100 on dev set.

| Method | MRR@100 | $\Delta$ vs. OPQ |
|---|---|---|
| OPQ (token-wise) | 0.352 | — |
| Token-wise cond. | 0.361 | +0.009 |
| PLAID | 0.368 | +0.016 |
| CrossQ | 0.374 | +0.022 |
| CrossQ + light FT | 0.379 | +0.027 |

CrossQ maintains gains on longer documents (+0.022 MRR@100 vs. OPQ), though the relative improvement is slightly smaller than on passages. This aligns with our failure analysis (Appendix G.6): longer documents have more diffuse winner-token distributions, reducing the benefit of cross-token capacity allocation. We note that the DeepSets mean-pooling aggregation may underweight critical evidence in very long documents; exploring attention-based context encoders for this regime is promising future work.

## J.4. Training Memory Overhead

Table 33 reports peak GPU memory during training at 4 B/token with batch size 32.

Assignment tensors and associated intermediates add approximately 8.2 GB in our implementation. This includes assignment logits, softmax probabilities, backward buffers and framework allocation overhead. The raw probability tensor alone, $\pi_{j,m,k} \in \mathbb{R}^{N \times M \times K}$, for 32 queries, 32 candidates per query, 180 document tokens, 4 additive stages and 256 codebook entries requires

$$32 \times 32 \times 180 \times 4 \times 256 \times 4 = 754{,}974{,}720 \text{ bytes}$$
$$\approx 0.75 \text{ GB}. \tag{85}$$

Thus, the reported 8.2 GB reflects the full assignment-related training footprint rather than the raw probability tensor alone. Overall, CrossQ training uses 40.0 GB peak memory, corresponding to a $40.0/28.3 \approx 1.41\times$ memory increase over PQ training. Memory scales approximately linearly with batch size and candidate count for memory-constrained settings gradient checkpointing or reduced candidate sets can be used with modest quality degradation.

# K. Discussion of Limitations

This section provides extended discussion and additional experiments addressing generalization, indexing overhead, related work, failure cases, STE stability and hyperparameter sensitivity.

## K.1. Generalization Beyond Max-Sim Scoring

CrossQ is designed for ColBERT-style max-sim scoring (Eq. 25). We discuss potential extensions to other late-interaction variants:

**Sum-of-max scoring.** Some variants score documents by summing document-side maxima: $s(Q, D) = \sum_j \max_i \langle q_i, d_j \rangle$. CrossQ's document context $h(D)$ could condition on which document tokens are likely to be selected, though the optimization signal would require corresponding adjustment to the listwise loss.

**Attention-based pooling.** For models using attention-weighted aggregation (e.g., Poly-encoders), the "winner" concept becomes soft attention weights. CrossQ could condition on attention entropy to allocate precision toward high-attention tokens, with the listwise loss naturally extending to attention-weighted scores.

*Table 33.* Training memory breakdown at 4 B/token with batch size 32 on one A100 80GB GPU.

| Component | Memory (GB) | % of Total |
|---|---|---|
| Retriever (frozen) | 1.8 | 4.5% |
| Document embeddings | 12.4 | 31.0% |
| Assignment tensors and intermediates | 8.2 | 20.5% |
| Codebooks & MLPs | 0.5 | 1.3% |
| Optimizer states | 6.1 | 15.3% |
| Activations & gradients | 11.0 | 27.4% |
| Total (CrossQ) | 40.0 | 100% |
| PQ baseline | 28.3 | – |

**Cross-encoder distillation.** When compressing for cross-encoder reranking, CrossQ's ranking-aligned training transfers directly the teacher simply becomes the cross-encoder. However, cross-encoders lack explicit token-level winners, so document context would target tokens with high gradient attribution instead.

**General recipe.** The core CrossQ principle *condition quantization on task-relevant structure* generalizes beyond max-sim:
1. Identify which tokens disproportionately affect the scoring operator.
2. Design conditioning signals that predict token importance without query access.
3. Train with losses aligned to the downstream ranking metric.
We leave empirical validation of these extensions to future work, as each requires operator-specific architectural choices and training signals.

### K.2. Backbone Generalization
To test whether CrossQ's mechanism transfers across retriever backbones, we evaluate on a publicly available ColBERTv2-family encoder at $d = 256$, keeping all other settings identical (matched footprint, candidate pool, evaluation protocol). The best baseline is the strongest non-CrossQ method at matched footprint.

*Table 34.* CrossQ gain transfers across retriever backbones (4 B/token, MS MARCO).

| Backbone / $d$ | Best baseline MRR@10 | CrossQ MRR@10 |
|---|---|---|
| BERT-base / 128 (main paper) | 0.381 | 0.386 |
| ColBERTv2-family / 256 | 0.401 | 0.406 |

The +0.005 MRR@10 gap is consistent across both backbones. While we do not claim exhaustive coverage of larger encoders or alternative tokenizers, this result suggests the mechanism transfers to higher-dimensional retrievers in the same family.

### K.3. Indexing Overhead at Scale
We provide concrete wall-clock estimates for large-scale deployment (Table 35).

*Table 35.* Projected indexing time at scale (single A100 GPU).

| Corpus Size | PQ | CrossQ | Overhead | Parallelized (8 GPU) |
|---|---|---|---|---|
| MS MARCO (8.8M) | 1.02 hr | 1.33 hr | +18 min | 10 min |
| Wikipedia (21M) | 2.4 hr | 3.1 hr | +42 min | 24 min |
| CC-News (100M) | 11.5 hr | 15.0 hr | +3.5 hr | 1.9 hr |
| Web-scale (1B) | 115 hr | 150 hr | +35 hr | 19 hr |

**Amortization analysis.** The additional 35 hours for 1B documents is a one-time offline cost. For corpora requiring frequent re-indexing (e.g., news), the overhead is more significant; we recommend incremental indexing strategies where only new documents are processed.

**Optimization opportunities.** CrossQ's context computation is embarrassingly parallel and amenable to: (i) batched GPU inference for $h(D)$, (ii) caching context encoders in mixed-precision and (iii) approximate context via locality-sensitive hashing. Further engineering, such as batched context inference, mixed-precision context encoders and approximate context computation, may reduce this overhead. We leave such systems optimization to future work.

### K.4. Comparison with Learned Compression Methods

We discuss related neural compression approaches not included in our main experiments:

**JPQ (Zhan et al., 2021)**  Joint Passage and Query encoding learns PQ codebooks end-to-end with the retriever. Key differences from CrossQ:
- JPQ targets single-vector (dense) retrieval, CrossQ targets multi-vector late interaction.
- JPQ jointly trains encoder + quantizer, CrossQ compresses a frozen encoder.
- JPQ optimizes contrastive loss, CrossQ uses listwise distillation + pairwise ranking.

JPQ's joint training could complement CrossQ; one could jointly fine-tune the encoder with CrossQ's ranking-aligned quantization, which our "light fine-tuning" variant partially explores.

**RepCONC (Zhan et al., 2022)**  Representation Compression with Contrastive learning also targets dense retrieval with learned quantization. Like JPQ, it operates on single-vector representations and is not directly applicable to multi-vector late interaction without architectural modification.

**BiDR (Xiao et al., 2022)**  Bi-directional Distillation for Retrieval distills cross-encoders into dense retrievers. This is orthogonal to CrossQ, BiDR could provide the teacher scores for CrossQ's listwise distillation.

**Why we omit direct comparison.**  These methods target fundamentally different retrieval architectures (single-vector vs. multi-vector). Adapting them to late interaction would require non-trivial modifications that risk unfair comparison. We instead compare against methods naturally applicable to multi-vector compression: PQ/OPQ applied per-token, QINCo2-style conditional quantization and PLAID's coupled compression-routing system.

### K.5. Extended Failure Case Analysis

We provide deeper analysis of when CrossQ underperforms and potential mitigations.

**Characterizing failure cases.**  Table 36 analyzes dataset characteristics where CrossQ's gains are smallest.

*Table 36.* Dataset characteristics vs. CrossQ improvement over PLAID (4 B/token).

| Dataset | Avg doc len | Winner entropy | $\Delta$nDCG@10 |
|---|---|---|---|
| HotpotQA | 92 | 1.82 | +1.1 |
| FiQA | 134 | 2.01 | +0.8 |
| NFCorpus | 181 | 2.34 | +0.8 |
| ArguAna | 248 | 2.89 | +1.4 |
| Touché-2020 | 312 | 3.12 | +0.8 |

Winner entropy measures the dispersion of max-sim winners across document tokens (higher = more diffuse). The relationship between winner entropy and CrossQ gains is suggestive but not strictly monotonic: Touché-2020 shows smaller gains under diffuse evidence, while ArguAna remains favorable despite high entropy. We therefore treat entropy as one diagnostic rather than a complete explanation of cross-dataset behavior.

**Attention pooling as selective mitigation.**  Attention pooling is a plausible mitigation for long or high-entropy documents, but selective hybrid variants were not part of our finalized evaluation protocol. We therefore leave a systematic DeepSets/attention context encoder study to future work.

**Fundamental limitation.**  When relevance is distributed uniformly across tokens (e.g., argumentative essays where every sentence contributes evidence), *any* precision allocation strategy has limited benefit the optimal solution approaches uniform allocation. CrossQ is most effective for corpora with sparse, localized evidence patterns.

### K.6. STE Stability Analysis

We provide training dynamics analysis for the straight-through estimator used at 2 B/token.

**Convergence consistency.**  Figure 11 (left) shows training loss curves for 3 seeds. All runs converge to similar final loss ($0.342 \pm 0.008$) with comparable trajectories. The variance band remains tight throughout training, indicating stable optimization despite STE's biased gradients.

**Gradient behavior.**  Figure 11 (middle) shows gradient norm distributions at training steps 5K, 10K and 20K. Gradient norms remain bounded (99th percentile $< 2.5$) with no explosion events across all seeds. The gradient clipping threshold (1.0) is triggered in $<3\%$ of steps, primarily during early training when $\tau_q$ is still high.

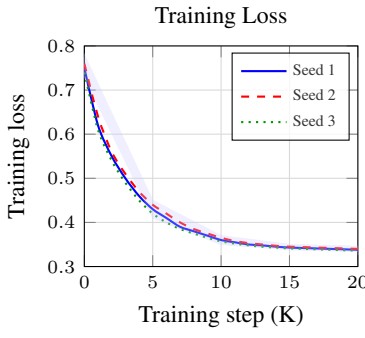 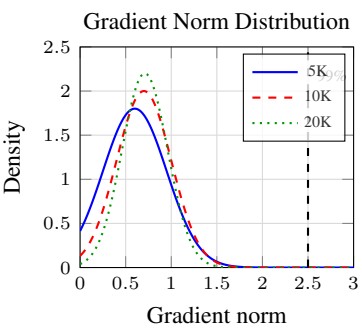 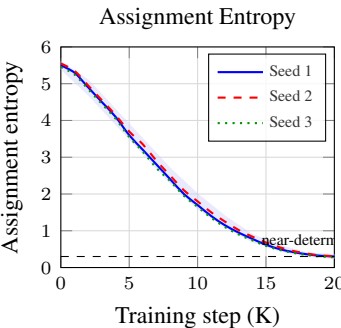

*Figure 11.* STE training stability across 3 random seeds (2 B/token). **Left:** Training loss curves converge to similar final values $(0.342 \pm 0.008)$ with tight variance bands. **Middle:** Gradient norm distributions at steps 5K, 10K and 20K remain bounded (99th percentile $< 2.5$) with no explosion events. **Right:** Assignment entropy decreases smoothly from $\sim 5.5$ (near-uniform) to $\sim 0.3$ (near-deterministic), confirming successful sharpening without collapse.

**Assignment sharpness.** Figure 11 (right) tracks assignment entropy $H(\pi_{j,m,\cdot})$ averaged over tokens. Entropy decreases smoothly from $\sim 5.5$ (near-uniform over 256 codes) to $\sim 0.3$ (near-deterministic), confirming that STE training successfully sharpens assignments without collapse.

**Quantitative stability metrics.** Table 37 reports stability metrics across seeds.

*Table 37.* STE training stability metrics (2 B/token, 3 seeds).

| Metric | Value (mean $\pm$ std) |
|---|---|
| Final training loss | $0.342 \pm 0.008$ |
| Final MRR@10 (dev) | $0.369 \pm 0.001$ |
| Gradient norm (median) | $0.71 \pm 0.03$ |
| Gradient clip frequency | $2.8\% \pm 0.4\%$ |
| Code utilization | $94.2\% \pm 1.1\%$ |

Code utilization (fraction of codebook entries used by $>0.1\%$ of tokens) confirms no mode collapse. The low variance across all metrics supports our claim of consistent STE convergence.

### K.7. Hyperparameter Sensitivity and Tuning Complexity

We analyze sensitivity to key hyperparameters and compare tuning burden against baselines.

**Critical vs. robust hyperparameters.** Table 38 categorizes hyperparameters by sensitivity.

*Table 38.* Hyperparameter sensitivity analysis (4 B/token).

| Hyperparameter | Range tested | MRR@10 range | Sensitivity |
|---|---|---|---|
| Listwise temp $\tau$ | $[0.02, 0.20]$ | $[0.378, 0.386]$ | Medium |
| Pairwise sharpness $\gamma$ | $[1, 8]$ | $[0.381, 0.386]$ | Low |
| Listwise weight $\alpha$ | $[0.5, 2.0]$ | $[0.382, 0.386]$ | Low |
| Pairwise weight $\beta$ | $[0, 1.0]$ | $[0.377, 0.386]$ | Medium |
| Recon weight $\lambda$ | $[0, 0.02]$ | $[0.384, 0.386]$ | Low |
| Temp decay $\eta$ | $[0.999, 0.9998]$ | $[0.383, 0.386]$ | Low |

Only $\tau$ and $\beta$ show meaningful sensitivity. The listwise temperature $\tau$ controls score distribution sharpness and requires matching to the score scale of the retriever. The pairwise weight $\beta$ balances ranking focus; $\beta = 0$ (no pairwise loss) underperforms but any $\beta \in [0.25, 1.0]$ works well.

**Recommended defaults.** For practitioners, we recommend starting with our selected configuration ($\tau = 0.05, \gamma = 4$, $\alpha = 1.0, \beta = 1.0, \lambda = 0.01$). This achieved within 0.003 MRR@10 of the best configuration across all budgets without per-budget tuning.

**Tuning burden comparison.** Table 39 compares tuning complexity.
CrossQ has more hyperparameters than reconstruction-based methods, but most are robust (Table 38). In practice, we found a 12-point grid over ($\tau, \beta$) sufficient, comparable to PLAID's tuning burden. The additional complexity is justified by

*Table 39.* Tuning complexity comparison.

| Method | # Hyperparameters | Grid size tested |
|--------|-------------------|------------------|
| PQ     | 2 (M, K)          | 6                |
| OPQ    | 3 (M, K, iterations) | 12            |
| PLAID  | 4 (M, K, routing thresh., centroids) | 24 |
| CrossQ | 6 ($\tau, \gamma, \alpha, \beta, \lambda, \eta$) | 48 |

CrossQ's consistent gains across budgets and datasets.

