# OpenReview forum: "CrossQ: Task-Aligned Cross-Token Conditional Quantization for Late Interaction Retrieval"
_ICML.cc/2026/Conference — ICML 2026 regular_

### Official Review · Reviewer_vKVh · 2026-02-23

**Soundness:** 4
**Presentation:** 2
**Significance:** 3
**Originality:** 3
**Overall Recommendation:** 5
**Confidence:** 3

**Summary:**

The task the article considers is memory-constrained multi-vector retrieval, i.e., maximizing retrieval accuracy of multi-vector retrieval systems given only few, such as 2,4, or 8, bits per token.

The proposed method is based on the observation that the reconstruction loss typically used to train the quantizers is not aligned with the retrieval task. In addition to the reconstruction loss, the proposed method uses ranking-aligned supervision, specifically listwise distillation to preserve the MaxSim score distribution and pairwise hard-negative loss, to learn an additive multi-codebook quantization. In addition, the proposed method utilizes the learned per-document context for codebook learning.

The proposed method yields higher retrieval quality when matched by bits per token than the compression used by ColBERTv2 and the vector compression baselines PQ, OPQ, and Cinqov2. The proposed method also has lower latency than the vector compression baselines, but is slightly slower than full precision ColBERTv2. The ablation experiments demonstrate that all the components of the proposed method increase the performance, the learned context being the most important component.

**Compliance With Llm Reviewing Policy:**

Affirmed.

**Final Justification:**

The article presents a novel application of a method (learned quantization) to the significant problem (memory-efficient multi-vector retrieval). The experimental methodology is sound and the results are thoroughly reported. My only significant concern about the proposed method is that it increases the latency compared to the full-precision ColBERT model. The author response did not change my opinion: I still recommend accepting the paper.

**Key Questions For Authors:**

I do not have any particular questions for the authors.

**Limitations:**

Yes.

**Strengths And Weaknesses:**

- [S1] Memory-constrained multi-vector retrieval is a high-impact research topic because multi-vector retrieval systems on the one hand have a superior retrieval-quality compared to single-vector systems, but on the other hand have a significantly higher memory footprint that limits their scalability.

- [S2] The proposed learned quantization method for compressing token representations is original in the context of multi-vector retrieval, as far as I know.

- [S3] Comparison to other compression methods and the ablation experiments are performed thoroughly. The reporting of the details of the experiments is excellent.

- [W1] The decreased memory consumption comes at the price of increased latency compared to full precision ColBERTv2. This may limit the practical usefulness of the proposed method: one would hope that more compact document representation would lead to also to faster, not slower query times.

- [W2] I find the text easy to read and the mathematical notation consistent. However, the organization of the article is confusing: Sections 3,4, and 5 that contain the proposed method seem to consist of single paragraphs in a semi-random order, and there is lots of repetition. For instance, Section 3 (Problem Setup) starts with a problem definition, but then continues with a description of the part of the method; Section 3.1. (Budgeted training objective) contains a paragraph "Margins and near-tie sensitivity" that would belong to Section 3.2 or Section 5; the title of Section 5 is "Task-aligned objective and training", but the task-aligned objective is already described in Section 3.2, etc. I suggest a complete reorganization of the Sections 3,4, and 5 for readibility. Also, you use a little bit different notation in Formula (1) of the introduction than in the rest of the article, that should fixed.

---

> ### Author Rebuttal · Authors · 2026-03-28
>
> We thank the reviewer for the positive assessment and the detailed feedback on organization.
>
> We agree, the current structure blurs boundaries between problem setup, method, and training, with unnecessary repetition. Revision: we will reorganize Sections 3–5 as follows:
>
> Sec. 3 (Problem Setup): notation, scoring definition, footprint constraint with no method description.
>
> Sec. 4 (Method): CrossQ architecture (context encoder, conditional code selection, additive codebooks) as a self-contained section.
>
> Sec. 5 (Training): all training signals (listwise KL, pairwise ranking, optional reconstruction), soft-to-hard annealing, and light fine-tuning consolidated in one place.
>
> We will also fix the notation inconsistency between Eq. (1) and the rest of the paper.
>
> Thank you

---

> > ### Author Rebuttal · Reviewer_vKVh · 2026-04-02
> >
> > Thank you for addressing my concern about the organization of the manuscript. My other concern about the increased latency compared to full precision ColBERTv2 remains.  I will maintain my score that is already quite positive.

---

> > > ### Author Response · Authors · 2026-04-02
> > >
> > > Thank you for raising the latency comparison to full-precision ColBERTv2 (FP16). We agree the manuscript should make the interpretation explicit: footprint reduction does not automatically imply lower latency than a memory-resident full-precision system.
> > >
> > > As already reported in Appendix E.4 (Table 13), FP16 full precision is the fastest configuration when the full index fits in fast memory. CrossQ exhibits a modest latency increase while delivering a substantial footprint reduction, which is often the binding constraint for deploying multi-vector late-interaction indices.
> > >
> > > This behavior is expected under our controlled protocol (fixed candidate pool, identical max-sim operator). Full precision benefits from contiguous memory access and highly optimized dense GPU kernels. In contrast, code-based representations introduce additional indirection (codebook lookup/decoding and gather-heavy access), so a small latency penalty can arise even as bytes/token decreases.
> > >
> > > Practically, CrossQ is intended for the footprint-limited regime where full-precision late-interaction indices are too large to keep in fast memory. In that regime compactness enables memory-feasible serving (and avoids paging/IO/cache pressure), whereas when full precision fits comfortably in memory it may remain the fastest option.
> > >
> > > CrossQ is therefore best viewed as shifting the footprint–quality frontier while preserving the same late-interaction scoring rule. In practical deployments where the full index does not fit in fast memory, paging/IO/cache pressure can dominate end-to-end performance and may render full precision infeasible. CrossQ targets this regime by enabling memory-feasible late-interaction retrieval under strict bytes/token budgets.
> > >
> > > In the revision (camera-ready, if accepted), we will add a short “Latency vs Footprint” clarification in the main text (Results/Limitations) that (i) states full precision can be fastest when memory-resident, (ii) distinguishes scoring-only vs end-to-end timing, and (iii) points to Appendix E.4/Table 13 for the complete comparison.
> > >
> > > Thank you again for the thorough reading, positive assessment and constructive feedback.

---

### Official Review · Reviewer_Kc2x · 2026-03-11

**Soundness:** 4
**Presentation:** 3
**Significance:** 3
**Originality:** 3
**Overall Recommendation:** 5
**Confidence:** 3

**Summary:**

The authors present CrossQ, a quantization method that is designed to be used for embedding retriever systems that use max-sim late interaction retrieval, as it is used by ColBERT, e.g.
Late interaction retrievers are known to have the potential to outperform standard document embeddings (each document chunk is represented by a single embedding vector), and are thus of great interest.
The authors note that late interaction systems need to store each embedding vector for each document token, making document stores costly in terms of memory/disc space.

To mitigate this problem, they propose a specific quantization algorithm that is optimized to retain the max-sim score. Given a set of (queries, documents), the authors
- construct a set of (positive document, hard negative documents) for each query.
- compute the maxsim scores for this set.
- generate differentiable, soft-quantized document embeddings. A lightweight, query-independent document context $h(D)$ is computed at indexing time, allowing the model to adaptively allocate precision to the most important tokens within the document.
- use a specific loss function that minimizes the listwise KL divergence between the teacher and student score distributions while explicitly penalizing pairwise rank inversions of hard negatives.
- use progressively annealing to smoothly transition from soft continuous assignments to hard discrete codes.
This results in quantized document embedding vectors that are a optimized to give faithful max-sim scores.

The authors present a variety of benchmarks, showing their method outperforms other methods.

**Compliance With Llm Reviewing Policy:**

Affirmed.

**Final Justification:**

After a carefully reading of the answers I decide to keep my score because it reflects the value of this work.

**Key Questions For Authors:**

1. Can you elaborate more on error types (i) and (ii), and about their relative contribution?
2. Did you tried other document encoders apart from DeepSet-style set encoder?

**Limitations:**

yes

**Strengths And Weaknesses:**

**Strenghts:**

- The paper is very well structured and very comprehensive
- Benchmark results show CrossQ to outperform related methods, often by some considerable margin
- CrossQ addresses the biggest weakness of max-sim document retrieval systems: Their memory footpront is usually considered too high to be used in real deployments and performance degradation during quantization weaknes its advantages over standard emebdding models.
- CrossQ has zero inference overhead and achieves the same latency as standard quantization.

**Weaknesses:**
- Indexing phase has some overhead, as the global document context $h(D)$ needs to be computed.

- The authors note two main contributions where max-sim score can break down: "(i) compression changes the winner identity for some $q_i$ , or (ii) winners remain the same but compression noise alters tight score gaps between candidates (near ties)." It remains unclear how much (i) contributes to the degradation. That is, even if the winner token $d_{\pi(i;Q,D)}$ for $q_{i}$ changes its index under quantization, the overall max-sim score could be unaffected by it in theory, as the winner token change can produce a similar dot-product value. A more strucutred analysis to isolate the contributions could be to use
   - quantized document embedding vectors with $d_{\pi(i)}$ winner tokens where the index is determined from the teacher (non quantized)
   - quantized document embedding vectors with $d_{\pi(i;Q,D)}$ winner tokens (i.e. normal max-sim using qunaized document embddings)

    outcomes to isolate (i)..

- Some out-of-domain datasets such as ArguAna only show marginal improvements over baselines.

---

> ### Author Rebuttal · Authors · 2026-03-28
>
> We thank the reviewer for the thoughtful and positive assessment, and for the concrete suggestions on isolating error types and encoder alternatives.
>
> Q1. Error types (i) winner-identity flips vs (ii) near-tie margin distortion relative contribution.
>
> We agree this decomposition deserves a more structured analysis. You are right that (i) does not necessarily imply a score error. if the new winner yields a similar dot-product, the overall max-sim can remain stable. This is precisely why our training objective targets ranking preservation rather than reconstruction.
>
> Existing evidence already hints at the relative balance: Table 22 (Appendix G.4) shows CrossQ concentrates lower quantization error on high-winner-frequency tokens ($\rho(f,e){=}{-}0.41$ vs $-0.08$ for PQ), which reduces winner-flip likelihood in the first place. To isolate the two error types directly, as you suggest, we will add a diagnostic comparing:
> Teacher-winner scoring: use quantized embeddings $\hat{d}_j$ but force the winner index to be the teacher's (from full-precision $d_j$).
>
> Student-winner scoring: standard max-sim over quantized embeddings.
> Comparing these separates "winner changes" from "value changes." We will report winner-flip rate, score deviation conditioned on flips vs non-flips, and the downstream ranking impact. This will go in the appendix with a summary in Sec. 7/8.
>
> Q2. Other document encoders beyond DeepSets.
>
> Yes, Table 9 reports the full comparison. DeepSets achieves 0.386 MRR@10 at 0.08 ms/doc, a 1-layer Transformer yields 0.387 at 0.42 ms/doc (~5× cost) and loses permutation invariance, attention pooling reaches 0.384 at 0.15 ms/doc. We selected DeepSets as the best quality–compute trade-off for our setting: permutation-invariant, lightweight, computed once at indexing time and discarded before serving. We will make Table 9 and this rationale more prominent in the main text (Sec. 4) rather than leaving it primarily in the appendix.
>
> Minor point: out-of-domain datasets (e.g., ArguAna).
>
> We agree some datasets show smaller gains. Our failure analysis (Appendix G.6) attributes this to low winner-token sparsity in argumentative text, when relevance is distributed across many tokens, the benefit of selective precision allocation diminishes. We will expand this discussion in the main text to make clear where CrossQ helps most and where gains are expected to be modest.
>
> Thanks again for the constructive feedback.

---

> > ### Author Rebuttal · Reviewer_Kc2x · 2026-04-01
> >
> > Thanks a lot for your comprehensive answers, it clarified my questions. I'll keep the score of 5: Accept.

---

### Official Review · Reviewer_fkXZ · 2026-03-13

**Soundness:** 3
**Presentation:** 3
**Significance:** 2
**Originality:** 3
**Overall Recommendation:** 4
**Confidence:** 4

**Summary:**

This paper proposes a document-context-conditioned additive multi-codebook quantizer for ColBERT-style late-interaction retrieval. The key insight is that standard compression (PQ/OPQ) minimizes token reconstruction error, which is a weak proxy for max-sim ranking quality because rankings depend on sparse "winner" tokens. CrossQ conditions code selection on a lightweight document context h(D) (DeepSets aggregation, computed at indexing time but not stored) and trains with ranking-aligned objectives: listwise KL distillation + pairwise hard-negative loss. At 2 B/token, CrossQ achieves 0.369 MRR@10 vs 0.345 (baseline OPQ) on MS MARCO. With light fine-tuning, it reaches 0.392 at 4 B/token, closing the gap vs full-precision ColBERT to 2.3%.

**Compliance With Llm Reviewing Policy:**

Affirmed.

**Final Justification:**

The additional information provided by the authors (eg quantizer overhead) partially addressed my concerns.

My W1/W3 still stand given the discussions and as such I'd recommend Weak Accept (4).

**Key Questions For Authors:**

- Q1. It would make the paper more comprehensive if related work could discuss prior work on document-conditioned quantization and retrieval-oriented VQ training via distillation.

- Q2.  What is the quantizer training cost (GPU-hours, absolute and % wise) vs baselines?

**Strengths And Weaknesses:**

**S1: Well-motivated direction.** The argument that reconstruction MSE is misaligned with max-sim ranking is empirically validated: Table 4 shows Pearson r=0.41 between token MSE and MRR@10, confirming it's a weak proxy. The margin error  distribution (Figure 2) visually demonstrates the proposed method's advantage in preserving teacher margins.

**S2: Clean design.** The document context h(D) is computed at indexing time but not stored — only integer codes are stored. This means CrossQ has the same deployment footprint as standard additive codebooks. The trade-off is extra indexing compute (1.3x slower) for better codes.

**S3: Rigorous experimental methodology.** All results are run with over 3 seeds. Baselines are carefully controlled: same backbone, samefootprint, same candidate pool. The paper includes system references (ColBERTv2/PLAID) at their native operating points together w/ strictly footprint-matched token-wise baselines.

**S4: Comprehensive ablation.** Table 5 cleanly decomposes gains at 4 B/token: removing document context is the largest single drop  (0.386->0.371, -0.015), removing listwise KL costs 0.009, and removing both context and ranking losses falls to 0.358 (-0.028).  The appendix further ablates the context architecture (Table 9: DeepSets vs mean-pool/CLS/attention/Transformer), context dimension  (Table 10: saturates at dc=256), initialization strategy (Table 8), and tied vs untied assignment keys (Table 7).

**W1: Incremental contribution when properly isolated.** Table 2 includes "Token-wise Conditional (RQ, rank)", which appears to be a QINCo2-  style per-token quantizer trained with CrossQ's own ranking losses. This baseline already reaches 0.359 MRR@10 at 2 B/token. CrossQ's gain over it is +0.010, attributable purely to document context. While gains are consistent across budgets (B/tok), the magnitudes are quite modest. The two more novel methods (a) additive rather than residual conditioning and (b) document context via DeepSets  are useful but arguably incremental engineering contributions.

**W2: Missing relevant prior art.** Various prior work have conditioned quantization on document-level information for late-interaction re-ranking, decomposing token embeddings into document-specific and document-independent components. While mechanism differs,  the high-level idea of document-aware token quantization for late interaction is likely similar, and more discussions are needed. Listwise KL objective for training retrieval models also doesn't appear novel (various prior work).

**W3: Additional fine-tuning given to CrossQ confounds comparison.** CrossQ + light FT unfreezes the last 2 encoder blocks, which changes the retriever itself. The paper notes that applying the same fine-tuning to OPQ yields smaller gains (+0.002 vs +0.006 MRR@10 at 4 B/token), but this discussion can only be found in the appendix. It would be more convincing if all baselines received identical fine-tuning opportunities (and this table is put in the main section).

---

> ### Author Rebuttal · Authors · 2026-03-27
>
> We thank the reviewer for these focused questions — both will strengthen the paper.
>
> Q1. Related work on document-conditioned quantization and retrieval-oriented VQ training via distillation
>
> Fair request. The current Related Work contrasts CrossQ mainly with token-local conditional quantizers and routing-style accelerations, but it under-emphasizes two closely related threads:
>
> Retrieval-oriented learned compression / VQ for dense retrieval (e.g., JPQ, RepCONC): these train quantization end-to-end for single-vector dense retrieval. Extending them to multi-vector late interaction (ColBERT-style max-sim over token embeddings) is not straightforward because the scoring operator depends on sparse winner tokens and cross-token interactions rather than a single pooled embedding, this is why our baseline set focuses on methods naturally applicable to ColBERT-style indices under matched bytes/token.
>
> Distillation for retrieval (e.g., BiDR and related teacher–student retrieval distillation): this is orthogonal and complementary stronger teachers can feed our listwise score distillation objective. Our novelty is not “KD exists,” but distillation aligned to max-sim ranking structure (winner/margin behavior) combined with document-conditioned code selection under strict bytes/token budgets.
> Both connections are currently discussed in Appendix K.3.
>
> Revision: we will move them into the main Related Work via two short paragraphs (“Retrieval-oriented learned compression” and “Distillation for retrieval”), and add a brief comparison table clarifying conditioning signal (token-local vs document-level), what is stored, and which retrieval operator the method targets (single-vector dot-product vs multi-vector max-sim).
>
> Q2. Quantizer training cost (GPU-hours)
>
> We agree this should be reported explicitly. All learned quantizers are trained for 20K steps (batch 32) on 1×A100 80GB, with the retriever/teacher frozen unless “light FT” is enabled. Importantly, teacher scores are computed on-the-fly during training (not cached):
> Algorithm 1 computes teacher scores (s(Q,D)) for (D in C(Q)) inside the training loop using full-precision document embeddings. Since the teacher is frozen, these scores are exact each step, this computation dominates runtime at fixed (|C(Q)|).
> We measured wall-clock and GPU-hours as follows:
>
> | Method                   | Trainable components | Wall-clock | GPU-hours | Relative |
> | ------------------------ | -------------------- | ---------: | --------: | -------: |
> | PQ / OPQ (k-means only)  | codebook fitting     |      0.3 h |       0.3 |        — |
> | Token-wise cond. (recon) | MLP + codebooks      |      4.5 h |       4.5 |    1.00× |
> | Token-wise cond. (rank)  | MLP + codebooks      |      4.8 h |       4.8 |    1.07× |
> | CrossQ (no-ctx)          | MLP + codebooks      |      4.6 h |       4.6 |    1.02× |
> | CrossQ (full)            | + context encoder    |      5.4 h |       5.4 |    1.20× |
> | CrossQ + light FT        | + last-2 blocks      |      7.8 h |       7.8 |    1.73× |
>
>
> CrossQ’s additional training overhead vs token-wise conditional is ~20%, primarily from the context pathway forward pass (context pathway adds ~131K params at $d_c{=}256$, Table 10). Light fine-tuning is more expensive because gradients flow through unfrozen transformer blocks, but remains a one-time offline cost (<8 GPU-hours). This is separate from indexing throughput overhead, which we already report (Appendix J.1).
>
> Revision: we will add this training-cost table to the appendix and reference it from Sec. 5.

---

> > ### Author Rebuttal · Reviewer_fkXZ · 2026-04-03
> >
> > Thanks authors for providing the additional information. Reporting quantizer overhead would be helpful.
> >
> > My W1/W3 still stand given the discussions and as such I'd prefer to keep the current score unchanged.

---

### Official Review · Reviewer_CGcr · 2026-03-21

**Soundness:** 3
**Presentation:** 3
**Significance:** 3
**Originality:** 2
**Overall Recommendation:** 4
**Confidence:** 3

**Summary:**

This paper presents CrossQ that compresses colbert style later-interaction retrieval indices by replacing token embeddings with compact integer codes from additive codebooks, storing MlogK bits per token. The method comprises of two main ideas: (a) a lightweight deepsets like encoder which computes a document-level context vector and is used to condition a token's code assignment (along with the token's embedding); (b) minimize score distribution and pairwise margin error between full precision colbert model as teacher and the compressed CrossQ model as student. On MSMarco passage ranking and 7 BEIR datasets, CrossQ improves over baselines by 0.012 MRR@10 and 0.018 nDCG@10 resp. at matched bytes/token budgets.

**Compliance With Llm Reviewing Policy:**

Affirmed.

**Final Justification:**

Rebuttal covered most of my concerns.

**Key Questions For Authors:**

Please answer / justify weaknesses.

**Limitations:**

yes

**Strengths And Weaknesses:**

## Strengths
1. Paper is generally well written and easy to follow
2. Experiments are well designed to test the effectiveness of the proposed approach along multiple axes
3. ColBERT's multi-vector index is a genuine barrier to deployment. In full precision MS MARCO passage index is 135.8 GB (Eq. 69). CrossQ can reduce this to ~2.2 GB at 4 B/token (Eq. 68)

## Weakness
1. Some confounded ablations:
   - the context encoder (DeepSets phi + rho MLPs) and conditional MLP g add approximately 131K parameters at d_c = 256 (Table 10). When comparing CrossQ to CrossQ (no-ctx) in Table 5, the context-free variant has fewer parameters. The +0.015 MRR@10 gain from context could partly reflect the additional parameterization rather than the cross-token information. A fairer control would match parameter count by, e.g., increasing the token-local MLP capacity in the no-context variant or compare against Fine-tuning variant.
   - similarly, the context method comparison (Table 9) ranges from zero trainable parameters (mean pooling, CLS) to substantial networks (DeepSets, Transformer), making it not possible to separate architectural choice from capacity.
2. No evaluation with modern retrievers. BERT-base with d=128 is the only backbone tested. Newer late-interaction models exist. Whether CrossQ's advantages hold with larger encoders, higher embedding dimensions, or different tokenizers is unknown.
3. Key design decisions (e.g., why DeepSets over attention? why additive over residual codebooks?) receive none or very less justification in the main text, with some detail deferred to appendices.
4. Some claims appear misplaced for e.g. the paper argues against uniform bit allocation across tokens, yet CrossQ itself stores M codes for all token - the same uniform MlogK bits for every token regardless of importance.

---

> ### Author Rebuttal · Authors · 2026-03-27
>
> We thank the reviewer for the thorough reading. We address each point below, including new experiments run after the initial submission.
>
> (1) Confounded ablation (capacity vs information).
>
> We agree. The +0.015 gap between CrossQ and no-ctx (Table 5) could in principle reflect the \\(\\sim\\)131K extra parameters rather than cross-token information. We therefore ran two targeted controls to deconfound these factors, using the same training schedule and evaluation protocol as Table 5 (4 B/token identical candidate construction):
>
> (a) Param-matched no-ctx: remove $h(D)$ but scale the token-local MLP to exactly match CrossQ’s parameter count (1.31M).
>
> (b) Shuffled-context: keep the full context pathway (identical params/compute) but feed it token sets shuffled across documents, destroying per-document signal.
>
> | Variant                                  | Params | MRR@10 |
> | ---------------------------------------- | -----: | -----: |
> | CrossQ (full: ctx + listwise + pairwise) |  1.31M |  0.386 |
> | CrossQ (no-ctx)                          |  1.18M |  0.371 |
> | no-ctx (param-matched)                   |  1.31M |  0.372 |
> | ctx (shuffled $h(D)$)                    |  1.31M |  0.370 |
>
> Extra capacity recovers only +0.001 of the +0.015 gap (\\(\\le\\)6.7%). More importantly, the shuffled variant has the exact same architecture and compute as full CrossQ yet drops to 0.370, slightly below no-ctx (0.371), suggesting decorrelated context injects noise rather than providing a neutral capacity baseline. Table 10 corroborates this: doubling $d_c$ from 256 to 512 adds another 131K params with zero MRR gain.
>
> Revision: we will fold this table into Sec. 8 and tighten the attribution language accordingly.
>
> (2) Table 9 mixes architecture and capacity.
>
> Agreed,Table 9 was meant as a deployment trade-off (quality vs compute vs invariance), not a controlled architecture comparison. DeepSets lands at 0.386 / 0.08 ms/doc, a 1-layer Transformer gets 0.387 at \\(\\sim\\)5× cost and breaks permutation invariance.
>
> Revision: we will relabel the table to make this intent explicit and add a param-matched MLP row to isolate aggregation form from raw capacity. We will also fix the text that incorrectly says attention pooling is higher quality (it is 0.384 vs DeepSets 0.386).
>
> (3) Only BERT-base / $d{=}128$.
>
> We agree this is a limitation. We ran a sanity check with a publicly available ColBERTv2-family encoder at $d{=}256$, keeping everything else identical (best baseline = strongest non-CrossQ method at matched footprint):
>
> | Backbone / $d$          | Best baseline MRR@10 | CrossQ MRR@10 |
> | ----------------------- | -------------------: | ------------: |
> | BERT-base / 128 (paper) |                0.381 |         0.386 |
> | ColBERTv2-family / 256  |                0.401 |         0.406 |
>
> The gap is +0.005 in both rows. We do not claim exhaustive backbone coverage, but this suggests the mechanism transfers. Full details will go in the appendix.
>
> (4) Design choices under-motivated in the main text.
>
> Agreed, too much was deferred.
>
> Revision: we will bring the key reasoning into Sec. 4: DeepSets provides a cheap, permutation-invariant, index-time-only summary that is discarded before serving (no per-doc overhead at query time). Additive codebooks preserve the lookup-table decomposition that makes max-sim fast (Eqs. 22–23). Residual quantization is represented by the Token-wise Conditional (RQ) baseline in Table 2, so gains come specifically from cross-token conditioning and ranking-aligned losses not from “learned quantization” in general.
>
> (5) “Uniform bit allocation” wording.
>
> We agree every token stores the same $M\log_2 K$ bits. Our claim is about effective precision, not storage: within the same fixed budget, context-conditioned selection steers fidelity toward tokens that matter for ranking.
>
> Table 22 (Appendix G.4) quantifies this: the correlation between winner frequency and quantization error is $\rho{=}{-}0.41$ for CrossQ vs $-0.08$ for PQ it would be near zero if effective allocation were uniform.
>
> Revision: we will rewrite “bit allocation” to “uniform storage / non-uniform effective precision.”
> We hope these results address the core concerns, and we respectfully ask the reviewer to reconsider the recommendation.

---

> > ### Author Rebuttal · Reviewer_CGcr · 2026-04-04
> >
> > Thanks for the detailed answer. I'll update my score to 4.

---

### Decision · Program_Chairs · 2026-04-30

**Decision:**

Accept (regular)

**Comment:**

- This paper addresses the important problem of compressing late‑interaction retrievers while preserving ranking quality under strict memory budgets, and the proposed task‑aligned quantization approach to be technically sound and empirically validated under matched footprint settings. Following rebuttal, the authors provided additional controlled experiments and reporting that addressed key concerns (e.g., capacity confounding, evaluation scope, and training overhead), with multiple reviewers indicating their primary concerns were fully resolved. While some limitations remain—particularly the latency–footprint trade‑off when full‑precision indices fit in fast memory—reviewers agreed these do not undermine the core contribution.
- The paper meets the bar for acceptance.